

# Summertime surface PM$_1$ aerosol composition and size by source region at the Lampedusa island in the central Mediterranean Sea

Marc D. Mallet[1,2,3], Barbara D'Anna[2,4], Aurélie Même[2,*], Maria Chiara Bove[5,6], Federico Cassola[5,7], Giandomenico Pace[8], Karine Desboeufs[1], Claudia Di Biagio[1], Jean-Francois Doussin[1], Michel Maille[1], Dario Massabò[5], Jean Sciare[9], Pascal Zapf[1], Alcide Giorgio di Sarra[8] and Paola Formenti[1]

1. LISA, CNRS UMR7583, Université Paris Est Créteil (UPEC), Université Paris Diderot (UPD), Institut Pierre Simon Laplace (IPSL), Créteil, France

2. IRCELYON, CNRS UMR 5652, Univ. Lyon1, Lyon, France

3. Centre National d'Etudes Spatiales (CNES), Toulouse, France

4. LCE, CNRS UMR 7376, Aix-Marseille Université, Marseille, France

5. Department of Physics & INFN, University of Genoa, Genoa, Italy

6. ARPAL Physical Agents and Air Pollution Sector, La Spezia, Italy

7. ARPAL CFMI-PC, Genoa, Italy (current affiliation)

8. Laboratory for Observations and Analyses of Earth and Climate, ENEA, Rome, Italy

9. The Cyprus Institute, Energy, Environment and Water Research Center, Nicosia, Cyprus

* now at Bruker

## Abstract

Measurements of aerosol composition and size distributions were taken during the summer of 2013 at the remote island of Lampedusa in the southern central Mediterranean Sea. These measurements were part of the ChArMEx/ADRIMED (Chemistry and Aerosol Mediterranean Experiment/Aerosol Direct Radiative Forcing on the Mediterranean Climate) framework and took place during the Special Observation Period 1a (SOP-1a) from 11 June until 5 July 2013.

From compact time-of-flight aerosol mass spectrometer (cToF-AMS) measurements in the size range below 1 μm in aerodynamic diameter (PM$_1$), particles were predominately comprised of ammonium and sulphate. On average, ammonium sulphate contributed 63% to the non-refractory PM$_1$ mass, followed by organics (33%). The organic aerosol was generally





very highly oxidised ($f_{44}$ values were typically between 0.25 and 0.26). The contribution of
ammonium sulphate was generally higher than organic aerosol in comparison to
measurements taken in the western Mediterranean but is consistent with studies undertaken
in the eastern basin.
Source apportionment of organics using a statistical (positive matrix factorisation) model
revealed four factors; a hydrocarbon-like organic aerosol (HOA), a methanesulfonic acid
related oxygenated organic aerosol (MSA-OOA), a more oxidised oxygenated organic aerosol
(MO-OOA) and a less oxidised oxygenated organic aerosol we label (LO-OOA). The MO-OOA
was the dominant factor for most of the campaign (53% of the $PM_1$ OA mass). It was well
correlated with $SO_4^{2-}$, highly oxidised, and generally more dominant during easterly air
masses originating from the eastern Mediterranean and central Europe. The LO-OOA factor
had a very similar composition to the MO-OOA factor, but was more prevalent during
westerly winds with air masses originating from the Atlantic Ocean, the western
Mediterranean, and in high altitudes over France and Spain from mistral winds. The MSA-
OOA factor contributed an average 12% to the $PM_1$ OA and was more dominant during the
mistral winds. The HOA, representing observed primary organic aerosol only contributed 8%
of the average $PM_1$ OA during the campaign.
Even though Lampedusa is one of the most remote sites in the Mediterranean, $PM_1$
concentrations (10 ± 5 µg m$^{-3}$) were comparable to those observed in coastal cities and sites
closer to continental Europe. Cleaner conditions corresponded to higher wind speeds.
Nucleation and growth of new aerosol particles was observed during periods of northwesterly
winds. From a climatology analysis from 1999 until 2012, these periods were much more
prevalent during the measurement campaign than during the preceding 13 years. These
results support previous findings that highlight the importance of different large-scale
synoptic conditions in determining the regional and local aerosol composition and oxidation
and also suggest that a non-polluted surface atmosphere over the Mediterranean is rare.

## 59    1. Introduction

The Mediterranean Sea is a unique marine environment, surrounded by mountain ranges and
high coastal human populations from Africa, Europe, and Asia, and the two largest deserts in
the world; Sahara Desert to the south and Arabian Desert to the East. It presents a diverse



and dynamic atmospheric composition and is projected to undergo significant changes in the
contribution of freshwater (Sanchez-Gomez et al., 2009), sea surface temperature and
precipitation (Mariotti et al., 2015) over the coming decades. The burning of fossil fuels,
including shipping pollution, in southern Europe and in large Mediterranean cities, as well as
natural sources of aerosol such as sea salt, forest fires and mineral dust provide a highly
complex and dynamic mixture of organic and inorganic aerosol and aerosol precursors in this
region (Lelieveld et al., 2002). Elevated aerosol loadings over the Mediterranean basin have
been attributed to the long-range transport of continental anthropogenic aerosols (Perrone
et al., 2013; Sciare et al., 2003; Sciare et al., 2008) and mineral dust transported from Africa
(Querol et al., 2009b; Koçak et al., 2007). Boundary layer observations in the eastern
Mediterranean have shown significant influence of long-range transported continental
pollution from southern and central Europe (Sciare et al., 2003). Furthermore, biomass
burning aerosol has frequently been observed over the basin, in particular the dry season in
summer when forest fires are more common (Pace et al., 2005). Long-ranged plumes from
north American fires have also been observed at high altitudes (Formenti et al., 2002; Ortiz-
Amezcua et al., 2014; Brocchi et al., 2018; Ancellet et al., 2016).
Previous long-term observations of the chemical composition of aerosol in the Mediterranean
have shown that $PM_{10}$ (particulate mass with aerodynamic diameter less than 10 µm) is
composed of secondary ammonium sulphate, primary and secondary organic aerosol from
fossil fuels or biogenic origins, with contributions from natural aerosols from the Sahara
Desert and sea spray (Bove et al., 2016; Koulouri et al., 2008; Schembari et al., 2014; Calzolai
et al., 2015). Mineral and sea salt contributions are significantly less in $PM_{2.5}$ particle fraction
(Querol et al., 2009a). Coarse mode particles contribute to the direct radiative effect over the
Mediterranean (Perrone and Bergamo, 2011; Meloni et al., 2006) and can also act as
condensation sinks for pollutants (Pikridas et al., 2012). Smaller (sub-micron) aerosol
particles, while also contributing efficiently to the total aerosol optical depth in this region
(Formenti et al., 2018), can also act as efficient cloud condensation nuclei and therefore have
influence on cloud formation, lifetime and precipitation (Haywood and Boucher, 2000).
Understanding the impact of different natural and anthropogenic sources on the regional
composition of the atmosphere is therefore important in our understanding of the influences
they have on the climate over the Mediterranean basin and surrounding regions. It is also





now widely recognised that aerosols contribute to adverse health effects in humans (World
Health Organization, 2016).
Consideration of both the local and regional meteorology are needed to characterise the
sources and aging of aerosols (Petit et al., 2017). The National Oceanic and Atmospheric
Administration's (NOAA) Hybrid Single-Particle Lagrangian Integrated Trajectory model
(HYSPLIT; Stein et al., 2015) and other trajectory models (e.g. FLEXPART; (Stohl et al., 2005)
have become a widely-used resources in atmospheric studies to compute the backwards or
forwards trajectories of air masses at any point on Earth. They can be useful for identifying
the possible origin of a particular episode associated with elevated concentrations of aerosols
or gases. Combined with in-situ measurements over longer time periods, they provide a more
holistic approach in understanding the link between local or region meteorology and
atmospheric composition (Schmale et al., 2013; Tadros et al., 2018; Zhou et al., 2016). This is
particularly useful for remote sites where local emissions are insignificant or infrequent.
Investigation of the aerosol physical and chemical properties can also help distinguish their
respective sources. Positive matrix factorisation (PMF; (Paatero, 1997; Paatero and Tapper,
1994) has proved to be a useful statistical tool in identifying aerosol sources or aging
processes of organics. The source apportionment of $PM_{2.5}$ and $PM_{10}$ over the Mediterranean,
from PMF method, has been investigated in recent works and showed a large spatial
variability in source contributions (Becagli et al., 2012; 2017; Calzolai et al., 2015; Amato et
al., 2016; Diapouli et al., 2017). The PMF approach was also used to study the aerosol source
and aging processes by utilising the complex nature of organic aerosol in the Mediterranean
(e.g. Hildebrandt et al., 2010, 2011; Bougiatioti et al., 2014; Minguillón et al., 2016; Arndt et
al., 2017; Michoud et al., 2017). This approach has become increasingly feasible with the
recent widespread implementation of instruments capable of providing real-time, high time-
and mass-resolved non-refractory aerosol composition, such as aerosol mass spectrometers
(Ulbrich et al., 2009). PMF models have shown to successfully resolve the bulk-composition
of sub-micron organic aerosol into the contributions from various primary sources (e.g.
biomass burning, fossil fuel burning, cooking aerosol), but can also reveal the contributions
and characteristics of secondary (SOA) organic aerosol (Zhang et al., 2011). Factors with
similar mass spectra are consistently observed, albeit with different contributions at
measuring sites all around the world. The most commonly observed primary organic aerosol



factors are hydrocarbon-like OA (HOA), usually from fossil fuel burning as well as biomass
burning (BBA) while SOA can be usually separated into at least two factors, with low-volatility
oxygenated OA (LV-OOA) and Semi-Volatile-OOA (SV-OOA) as common examples (Zhang et
al., 2011; Crippa et al., 2014). Other types of OOA have also been observed, such as "Marine-
OOA" (Schmale et al., 2013), although these are more difficult to resolve given the shift
towards more uniform OA composition with aging.
The Chemistry-Aerosol Mediterranean Experiment (ChArMEx) collaborative research
program, and the Aerosol Direct Radiative Impact on the regional climate in the
MEDiterranean region (ADRIMED) project within, were undertaken to investigate the
chemistry and climate interactions within the Mediterranean (Mallet et al., 2016). From 11
June until 5 July 2013, numerous experimental setups were deployed across the western and
central Mediterranean in what is called the "Special Observation Period - 1a" (SOP-1a),
including intensive airborne measurements (Denjean et al., 2016). Two super-sites were set-
up at Ersa (at the northern tip of Corsica Island, France) and at the Lampedusa Island (Italy),
approximately 1000 km apart on a north-to-south axis in order to characterise surface aerosol
chemical, physical and optical properties (Mallet et al., 2016). Numerous secondary sites were
established along the Mediterranean coasts in Spain, Italy and Corsica beyond the SOP-1a
have also provided valuable knowledge of the atmospheric composition in the western and
central Mediterranean regions (Chrit et al., 2017; Chrit et al., 2018; Becagli et al., 2017).
In this paper, we present the first detailed characterisation of $PM_1$ in the central
Mediterranean region from measurements of size-resolved chemical composition from the
island site of Lampedusa during the ChArMex/ADRIMED SOP-1a. We investigate the source
apportionment of $PM_1$ by considering their chemical and microphysical properties along with
ancillary $PM_{10}$, gaseous and meteorological data, air mass back trajectories as well as
complimentary data collected at the Ersa site in Corsica.

## 2. Experimental

### 2.1 Sampling sites

Observations took place at the Roberto Serao station on the island of Lampedusa (35°31'5''N,
12°37'51'' E, 20 m above sea level) from 11 June to 5 July 2013. Ancillary measurements for



this study taken at Ersa at the northern tip of Cape Corsica (42°58'5" N, 9°22'49" E, 560 m
above sea level), are also considered. The position of the stations is shown in Figure 1.

## 2.2 Instrumentation, measurements and data

Instruments at the Lampedusa super-site were housed in the PEGASUS (Portable Gas Field
and Aerosol Sampling Unit) station, a portable observatory initiated by LISA, described in
Mallet et al. (2016). Relevant to this study, a c-ToF-AMS (Aerodyne Inc., Billerica, USA), was
used to measure the size-resolved composition of non-refractory particulate matter below 1
μm (NF-PM$_1$)(Drewnick et al., 2005). Data was collected with a 3-minute time resolution. The
c-ToF-AMS was operated from a certified Total Suspected Particulate (TPS) sampling head
(Rupprecht and Patashnick, Albany, NY, USA) followed by a cyclone impactor cutting off
aerosol particles larger than 1 μm in aerodynamic diameter.
A particle-into-liquid sampler (Metrohm PILS; Orsini et al. (2003)) was installed on a TSP inlet
and collected samples approximately every hour. Denuders to remove acid/base gases were
not used. Samples were analysed for major and organic anions (F$^-$, Cl$^-$, NO$_3^-$, SO$_4^{2-}$, PO$_4^{3-}$,
HCOO$^-$, CH$_3$COO$^-$, (COO$^-$)$_2$) and cations (Na$^+$, NH$_4^+$, K$^+$, Ca$^{2+}$, Mg$^{2+}$) using Ion Chromatography
(Metrohm, model 850 Professional IC) equipped with Metrosep A supp 7 pre-column and
column for anions measurements and Metrosep C4-250 mm pre-column and column for
cations measurements, and a 500 μL injection loop. The device was operated with a 1-hour
time resolution.
A 13-stage rotating cascade impactor nanoMOUDI (Model 125B, Marple et al., 1991) was used
to measure the size-segregated inorganic elemental composition. The nanoMoudi impactor,
also operated from the TSP inlet, allows the separation of the particles in 13 size classes from
10 nm to 10 μm diameter with a backup stage. Each sample was collected for 3 days to ensure
enough material was collected on each impactor stage. Filters were then analysed using X-ray
fluorescence (PW-2404 spectrometer by PANalytical™) for the particulate elemental
concentrations for elements from Na to Pb as described in Denjean et al. (2016))
A Scanning Mobility Particle Sizer (SMPS) measured the mobility number size distribution of
aerosols every 3 minutes from 14.6 to 661.2 nm. diameter. The instrument is composed by
an X-ray electrostatic classifier (TSI Inc., model 3080) and a differential Mobility Analyser
(DMA; TSI Inc., model 3081), and a condensation particle counter (CPC; TSI Inc., model 3775)



operated at 1.5/0.3 L min$^{-1}$ aerosol/sheath flows. Data were corrected to take into account
the particle electrical charging probabilities, the CPC counting efficiency, and diffusion losses.
Each scan was recorded with a 5-minute time resolution. A drier was not used on the SMPS
inlet and therefore the size distributions reported are for ambient conditions.
A GRIMM optical particle counter (OPC; GRIMM Inc., model 1.109) was used to measure the
number size distribution over 31 size classes ranging from 0.26 μm up to 32 μm (nominal
diameter range assuming the aerosol refractive index of latex spheres in the calibration
protocol). The instrument was operated at a 6-second resolution and data were acquired as
3-minute averages.
The equivalent black carbon mass concentration (eBC) was determined by the measurement
of light-attenuation at 880 nm performed by a spectral aethalometer (Magee Sci. model
AE31) operated at a 2-minute time resolution and equipped with a TSP particle inlet. As the
evaluation of eBC is used as a qualitative tracer of pollution, the factory mass conversion
factor of 16.6 m$^2$ g$^{-1}$ was applied to the raw measurement of attenuation without further
corrections.
The meteorological measurements (air pressure, temperature, relative humidity, wind
direction and speed and precipitation) were collected by a Vaisala Milos 500 station with a
sampling rate of 10 minutes. The wind sensor was installed on a 10-m meteorological tower,
while the air temperature and humidity were measured at a height of 2 m.
### 2.3. Data analysis
#### 2.3.1. Analysis of the cToF-AMS data
The cToF-AMS data set was processed using two different software analysis tools. The first
makes use of the widely-used and standard Igor Pro package, Squirrel (version 1.57G). This
software processes the raw data and analyzes the unit-mass resolution (UMR) output with a
fragmentation table reported in Aiken et al. (2008). The second method uses a cumulative
peak fitting analysis and residual analysis and allows the separation of multiple isobaric peaks
not taken into account in the traditional analysis of unit mass resolution squirrel data
treatment (Muller et al., 2011). Uncertainties in the major chemical species from the cToF-
AMS are typically of the order of ±20% (Drewnick et al., 2005).





The PM$_1$ sea salt concentration was estimated in the cTof-AMS by applying a scaling factor of
102 to the ion fragment at 57.98 assigned to NaCl as proposed by (Ovadnevaite et al., 2012).
The sea salt-SO$_4^{2-}$ (ss-SO$_4^{2-}$) was calculated as 0.252 * 0.3 * [seasalt], where 0.252 is the mass
ratio of SO$_4^{2-}$ to Na$^+$ in sea salt and 0.3 is the mass ratio of Na$^+$ to sea salt (Ghahremaninezhad
et al., 2016).
Unconstrained positive matrix factorisation was performed on both the unit-mass-resolution
spectra of organic aerosol as well as the peak-fitted peaks identified as organics using PMF2
v2.08D. This method requires both a matrix for both the organic signals as well as the errors
associated with the organics. For the peak-fitted signals, errors for each mass were estimated
as
$$\frac{\Delta I}{I} = \sqrt{(\alpha^2 t + (\beta + 1)I}$$

Where I is the ion signal, ΔI is the absolute uncertainty in the ion signal, α and β are constants
(1.2 and 0.001, respectively) and t is the instrumental sampling time in seconds (Drewnick et
al., 2009; Allan et al., 2003). For both UMR and peak-fitted inputs, up to 8 factors were
investigated by altering the seeds from 0 to 50 in increments of 1 and the fpeaks from -1 to 1
in increments of 0.1. Where I is the ion signal, ΔI is the absolute uncertainty in the ion signal,
α and β are constants (1.2 and 0.001, respectively) and t is the instrumental sampling time in
seconds. For both UMR and peak-fitted inputs, up to 8 factors were investigated by altering
the seeds from 0 to 50 in increments of 1 and the fpeaks from -1 to 1 in increments of 0.1.
This approach is explained in Ulbrich et al. (2009)
2.3.2. Air mass back-trajectory calculation and cluster analysis
In order to determine potential source regions for aerosols measured at Lampedusa during
the SOP-1a, a series of cluster analyses were performed on HYSPLIT air-mass back-trajectories
as per the following. Weekly GDAS1-analysis (Global Data Assimilation System; 1° resolution)
trajectory files were downloaded from the Air Resources Laboratory (ARL) of the National
Oceanic and Atmospheric Administration (NOAA) archive. 144-hour air-mass backwards
trajectories were then calculated every hour over the measurement period with ending point
at Lampedusa (height of 45 m) using HYSPLIT (Stein et al., 2015) from within the R-package,
SplitR. Cluster analyses were then performed on these calculated trajectories, using a
trajectory clustering function within the R-package, OpenAir (Carslaw and Ropkins, 2012).



Clustering was done using two different methods to calculate the similarity between different
trajectories. The first uses the Euclidean distance between the latitude and longitude of each
trajectory point (a total of 144 in this case, representing each hour prior to the arrival at the
receptor site). The second uses the similarity of the angles of each trajectory from the origin.
These two methods are described in Sirois and Bottenheim (1995). For each clustering
method, the number of clusters was altered from two up to ten. Six clusters identified using
the Euclidean-distance method were selected, producing a realistic separation of the air-mass
backwards trajectories and distinct and physically meaningful differences in aerosol
composition and size. An additional clustering analysis was also performed over 3 and 6 hour
intervals and using 96-hour backwards trajectories and yielded similar results.

## 3. Results and Discussion

### 3.1. Analysis of local and synoptic meteorology

The analysis of the hourly resolved 144-h air mass backwards trajectories provides an
indication of the origin of the air masses sampled at Lampedusa during the field campaign.
Six distinct clusters are identified (Figure 2). Cluster 1, "Eastern Mediterranean", is
representative of air masses that circulate around the eastern-central Mediterranean basin
before arriving at the Lampedusa site (average altitude of 400 m). Cluster 2, "Central Europe"
is representative of air masses arriving from central Europe (average altitude of 800 m).
Cluster 3, "Atlantic" is representative of more marine-like air masses that predominately
originate over the Atlantic Ocean, pass over the Strait of Gibraltar between Spain and
Morocco, and cross the western Mediterranean basin (average altitude of 500 m). Cluster 4,
"Western Europe", cluster 5, "Mistral (low)", and cluster 6, "Mistral (high)", all have similar
angular trajectories, but are distinguishable by their different wind speeds and altitudes
(although the Euclidian method of cluster analysis only considers differences in horizontal
distances). The two "Mistral" clusters typically originate over the northern Atlantic Ocean,
travel over France at a high altitude before descending over the western Mediterranean and
travelling with relatively higher wind speeds towards Lampedusa. The altitude of "Mistral
(high)" was, on average, higher than "Mistral (low)" (1400 m and 1000 m, respectively) and
also coincided with higher wind speeds at Lampedusa (13 ms$^{-1}$ and 9 ms$^{-1}$, respectively). In
comparison, the trajectories of the "Western Europe" cluster spent much more time



circulating at lower altitudes (700 m, on average) over the western Mediterranean basin and,
to a certain extent, east of the Lampedusa site.
As a complement, the pressure, temperature, relative humidity, wind speed and direction
time series recorded at the station are shown in Figure 3. Two main weather regimes are
observed: the former characterized by intense (up to nearly 20 m s$^{-1}$) northwesterly winds,
persisting for several days (10-13 June and 22-30 June) and cool temperatures, whereas the
latter  associated with low-gradient anticyclonic conditions and light winds from the east or
southeast, also favouring warmer temperatures (14-21 June and 1-3 July). Temperatures
were relatively stable over the sampling period, fluctuating between approximately 18.5 °C
and 28.2 °C. The relative humidity typically ranged from between 70% and 82% with very few
and very brief episodes of drier air masses (relative humidity close to or below 50%).
The wind speed and direction distributions during the campaign can be compared to the June-
July climatology from 1999 to 2012 (Figure 4). During the sampling period of this study, the
frequency of winds from the north-westerly sectors were nearly double the average when
compared to normal conditions, approaching 40% with high winds speeds exceeding 10 m s$^{-1}$
$^{1}$ observed during more than 20% of the time.
Data of sea level pressure and 1000 mbar meridional wind component composite anomalies
obtained from the National Center for Environmental Prediction (NCEP)/National Center for
Atmospheric Research (NCAR) Reanalysis (Kalnay et al. 1996) indicate that this particular
situation was induced by a "dipolar" pattern, characterized by positive pressure anomalies in
the Western Mediterranean and negative ones in the eastern part of the basin (see
Supplementary Figure S1). This produced a persisting, stronger than normal gradient over
Southern Italy. As a consequence, surface dust episodes typically driven by strong south or
southeasterly winds, associated to cyclonic systems moving along northern African coasts,
were basically absent during the campaign.
### 3.2 Aerosol composition
The dry NR-PM$_1$ concentrations measured at Lampedusa by the cTof-AMS ranged from 1.9 to
33.4 µg m$^{-3}$, with a mean of 10.2 µg m$^{-3}$ over the sampling period. Sulphate contributed the
most to the measured NR-PM$_1$ mass (41% ± 9% on average) followed by significant
contributions from organics (31% ± 8%) and ammonium (17% ± 3%). The eBC, nitrate and sea
salt (scaled from the NaCl component of m/z 58) contributed 6% (±4%), 1% (±0.4%) and 3%



(±2%), respectively. Figure 5 shows the total $PM_1$ concentration (calculated as the sum of the
individually measured species), with contribution from each of the species, as well as the
calculated $PM_1$ mass concentration from the SMPS (assuming an average density based on
the composition data). There was reasonable agreement between the $PM_1$ mass
concentration calculated from composition measurements and the SMPS, with discrepancy
observed during periods of high sulphate concentrations from the eastern Mediterranean,
likely due to the broad accumulation mode exceeding the upper size limits of the cToF-AMS
inlet. This figure also contains an indication of the air mass origin over the sampling period.
For most of the campaign there was a good agreement between the $PM_1$ $SO_4^{2+}$ and the TSP
$SO_4^{2+}$ concentration, with the exception of periods of high sea salt concentrations when the
TSP $SO_4^{2+}$ were significantly higher (see Supplementary Figure S2. Supporting measurements
of the size-segregated composition from the cascade impactor corroborate this, indicating a
higher contribution of elemental sulphur in the coarse mode during periods of higher sea salt
(Supplementary Figure S3). These periods corresponded with the "Mistral" air masses,
characterised by higher wind speeds and indicate the role of coarse mode particles in acting
as a condensation sink for sulphate species. This indicates that, in these circumstances, the
sea salt particles acted as a condensation sink for sulphate precursors. This has important
implications for the radiative properties of these aerosols by altering the scattering properties
and, potentially cloud condensation nuclei concentrations and composition.
Figure 6 shows the organic mass, split according to different OA factors from the PMF of the
OA peaks. From the unconstrained PMF of the UMR and peak-fitted organic mass spectra, the
most meaningful solution was found from a 4-factor solution of the UMR analysis (see
Supplementary Figures S4 and S5 for the mass reconstruction and time series' residuals). This
has resulted in one factor resembling to a primary organic aerosol and three oxygenated
organic aerosol (OOA) factors. Herein we label these factors HOA (hydrocarbon-like OA), MO-
OOA (more oxidised OOA), LO-OOA (less oxidised OOA) and MSA-OOA (methanesulfonic acid-
related OOA).
These factors were compared with ambient organic mass spectra listed in the AMS Spectral
Database (Ulbrich et al., 2009). The spectra for the OOA factors (see Figure 7) were strongly
correlated with each other (R > 0.97) and all three were similar to a "continental" OOA factor
observed in a ship campaign in the Artic (Chang et al., 2011), as well as a low-volatility OOA





factor identified in Paris (Crippa et al., 2013). Despite the similarities in their mass spectra,
they exhibited different diurnal trends (Figure 7) and time series that were associated with
different wind directions and air masses and were therefore not recombined into a single
OOA factor.
The MO-OOA factor was the most dominant factor during the field campaign (~53% of the
total OA mass) and was typical of low-volatile/highly oxidized OOA observed in many other
studies, in the Mediterranean, with high contributions of m/z 44 ($f_{44}$ = 0.31) (e.g. (Hildebrandt
et al., 2010; Hildebrandt et al., 2011). This factor was the most prominent during air masses
from the Eastern Mediterranean, central Europe and Atlantic (contributing to 71%, 58% and
55% to OA, respectively) and was strongly correlated with ammonium sulphate ($R^2$ = 0.68)
over the whole campaign. It also had a distinct diurnal trend, with concentrations increasing
during daylight hours, indicative of photochemical processing. The LO-OOA factor was
slightly less oxygenated ($f_{44}$ = 0.26) and exhibited a different time series, related to different
air masses, than the MO-OOA. The less oxygenated OOA factor has also been associated with
semi-volatile species and is often labelled as SV-OOA (Jimenez et al., 2009). Despite a distinct
diurnal profile similar to previously reported SV-OOA factors (with a peak in the early
morning), we refrain from labelling our LO-OOA this way because the mass spectrum was
generally much more oxygenated and contained less $f_{43}$ than typically reported SV-OOA (see
Figure 8).
The MSA-OOA factor contributed approximately 12% of the total OA during the campaign and
is likely related to the biogenic emission and processing of dimethyl sulfide (DMS) from
phytoplankton in the Mediterranean. This factor was also highly oxygenated ($f_{44}$ = 0.20), but
contains key peaks related to the fragmentation of MSA from the electron impact of the cToF-
AMS. The most prominent of these peaks were m/z 96 ($CH_4SO_3^+$),79 ($CH_3SO_2^+$), 78 ($CH_2SO_2^+$),
65 ($HSO_2^+$) and 45 ($CHS^+$). A similar factor was observed at Bird Island in the south Atlantic
Ocean (Schmale et al., 2015), albeit without the contribution of significant m/z 44, suggesting
a more aged or mixed aerosol during this campaign. A distinct diurnal pattern for the MSA-
OOA was not observed.
While the MO-OOA, LO-OOA and MSA-OOA factors represent secondary organic aerosol and
were the most dominant contribution of OA during the campaign (92% on average), a primary
organic aerosol factor was observed and identified here as hydrocarbon-like OA (HOA). The
mass spectrum of the HOA factor was characteristic of spectra observed in other studies, with





prominent peaks at m/z 95, 91, 83, 81, 71, 69, 57, 55, 43 and 41. Although HOA is typically
associated with emissions from incomplete combustion (Zhang et al., 2005), it was not well
correlated with other expected tracers such as eBC, CO and NOx. This HOA was typically
associated with south-westerly winds of low speed (<5 m s$^{-1}$; see Figure 9) and peaked at
approximately 6 am local time each morning. The poor correlation between the HOA factor
and eBC could have been due to a variety of local sources with different HOA and eBC
emission factors, a mixing of the PMF factor with some small peaks not associated with
combustion processes or from regional HOA that has undergone some transport without
significant oxidation. The signal fraction of each m/z of the mass spectrum of the HOA factor
however had strong correlations (0.69 < R < 0.89) with numerous hydrocarbon-like organic
aerosol (HOA) factors from other studies (Hersey et al., 2011; Ulbrich et al., 2009; Ng et al.,
2011; Lanz et al., 2007; Zhang et al., 2005) .

3.3 Comparison with other observations around the Mediterranean
There are many factors that could influence the composition, the concentration, and
oxidation level of different aerosol species over the Mediterranean. These include different
aerosol sources which can follow different seasonal or yearly trends (e.g. biogenic emissions)
as well as the existing aerosol load and the meteorological conditions that drive transport,
dilution and aging processes. Table 1 summarises the recent observations of NR-PM$_1$
composition from measurements in the remote Mediterranean.
The majority of previous studies of detailed PM$_1$ aerosol composition have been taken at
coastal sites around the Mediterranean (Mohr et al., 2012; Minguillón et al., 2016; Minguillón
et al., 2015; Haddad et al., 2013; Bozzetti et al., 2017) which could be expected to observe
higher concentrations than at Lampedusa due to proximity to sources (e.g. traffic, fossil fuel
use, heating, biomass burning, industrial activities). Aside from Lampedusa and the
observations presented in this study, measurements at Finokalia and Cape Corsica could be
considered the most remote sites where these measurements have been taken.
The PM$_1$ mass loading observed at Lampedusa is comparable to most of these other studies
performed at both remote marine sites and coastal sites (see Supplementary Table S1 for a
comparison with coastal urban sites). With the exception of Eastern Mediterranean, OA was
the dominant NR-PM$_1$ constituent and summertime OA was generally considered mostly



secondary, comprised of SV-OOA and LV-OOA with small contributions of HOA. For remote
sites, the results are consistent with a predominance of OA in $PM_1$ fraction in summer. PMF
analysis of Q-AMS measurements at the Finokalia remote site in the eastern Mediterranean
in the summer of 2008 showed two OOA factors and a distinct lack of HOA (Hildebrandt et
al., 2010). A more recent study during the late 2012 summer at Finokalia observed periods
influenced by biomass burning, but otherwise also observed mostly oxygenated organic
aerosol (Bougiatioti et al., 2014). Measurements undertaken in the western Mediterranean
at Cape Corse from 11 June until 6 August 2013, encompassed the sampling period of this
study. For the period from 15 July until 5 August, PMF analysis showed 55%, 27% and 13%
contributions of organic matter, sulphate and ammonium to non-refractory $PM_1$ (Michoud et
al., 2017). Secondary oxygenated VOCs dominated the VOC spectrum during the campaign
and were very well correlated with submicron organic aerosol. PMF analysis on the OA
revealed a 3-factor solution where SV-OOA and LV-OOA were dominant, contributing by 44%
and 53%, respectively, with a 4% HOA contribution. From the same measurements but
reported over the extended period from 11 June 11 until 5 August 5, there was a higher LV-
OOA contribution (62%)(Arndt et al., 2017) which is in agreement with our observations of
MO-OOA at Lampedusa. The OA was mostly portioned into MO-OOA and LO-OOA (81%),
indicative of well-aged or oxidised secondary organic aerosol from long-range transport of
pollutants. Furthermore, the contribution of ammonium sulphate was higher in this study
than of all those undertaken in the eastern Mediterranean basin, highlighting the role
contribution of sulphates across the Mediterranean.
Figure 8 displays the behavior of the $f_{44}$ and $f_{43}$ fragments obtained during the field campaign.
$f_{44}$,a proxy for OA oxidation (Jimenez et al., 2009), is calculated as the ratio of the mass at m/z
44 (mostly $CO_2^+$) to the total OA, while $f_{43}$, equal to the ratio of m/z 43 (mostly $C_2H_3O^+$) to the
total OA, typically represents less aged OA. $f_{44}$ was ~0.26 for the majority of the sampling
period (Q1; 0.25, Q3: 0.27), while $f_{43}$ was 0.036 (Q1: 0.028, Q3: 0.041). The campaign values
are compared to values from the spectra for the four PMF factors and those observed in other
field campaigns in the remote Mediterranean. The dotted lines (the so-called "Ng triangle"),
encapsulate the $f_{44}$ and $f_{43}$ values of atmospheric OA from a vast number of studies (Ng et al.,
2010), with the most aged OA in the top left corner and the most fresh in the bottom right.



The high $f_{44}$ values and the dominance of the highly oxygenated MO-OOA and LO-OOA factors
show that the organic aerosol was extremely aged compared to other measurements.
3.4 Links to meteorology
The contribution of the major submicron chemical species and OA sources is further explained
in the following by linking the measured and apportioned concentrations to the local
meteorology (i.e., wind speed and wind direction) and to the air mass back trajectories to
account for the long-range transport of aerosol as well as more distant sources. The bivariate
polar plots of these $PM_1$ species and $f_{44}$ as a function of wind speed and direction are shown
in Figure 9.
Considering that the sampling site on Lampedusa is on the north east tip of the island, it is
evident that the $SO_4^{2-}$ and $NH_4^+$ were likely a result of north-westerly marine air masses, in
agreement with previous results (e.g., Bove et al., 2016). Sea salt concentrations were highest
during high north-westerly wind speeds. Higher concentrations of $NO_3^-$, HOA and some of the
periods with elevated eBC concentrations were observed during low speed south-westerly
winds, likely a result of the human settlements and activity on the island of Lampedusa
(population of ~6000 located to the south west of the sampling site). Besides, the polar plot
for eBC showed a patchier pattern, indicative of more local or point sources and the elevated
signals were likely due to air masses passing over ship plumes. Although the mass spectra for
the LO-OOA and MO-OOA factors were very similar, their bivariate polar plots indicate
different sources or photochemical processes. The MO-OOA was more prominent during
north-easterly winds, indicating the most aged organics were influenced by air masses from
the eastern Mediterranean, either from long-ranged transport of from circulation of closer
pollution sources, while the LO-OOA was more dominant during northwesterly wind
directions and air masses from over the western Mediterranean. Figure 10 shows the average
contribution of each species during different air mass periods (see Supplementary Table S2
for the mean concentrations and standard deviations).
The highest concentrations of $PM_1$ were observed during "Eastern Mediterranean" and
"Central Europe" air mass periods, when significant lifetime over the lower altitude marine
environment and/or higher $SO_2$ emissions allowed the conversion and condensation of
sulphate. These aged aerosols are corroborated by the high number concentrations within
the accumulation mode during these periods relative to other periods, measured by the SMPS



as well as the size-resolved sulphate composition, as discussed in the next section⁻. In contrast
to the "Eastern Mediterranean" and "Central Europe" air masses, sulphate concentrations
were relatively low during the two "Mistral" air masses. This behavior has been found also in
$PM_{10}$, with elevated values of sea salt aerosol and low non-sea salt sulphate during Mistral
events (Becagli et al., 2017). The organic mass concentration was relatively uniform across
the periods of different air mass origins, with the exception of the high Mistral winds which
yielded OA concentrations approximately half the rest of the campaign and a higher
contribution of MSA-OOA in comparison with other periods.
### 3.5 Aerosol size distributions
There are distinctions between the measured $PM_1$ size distributions during periods of
different air mass origins (Figure 11). It should be noted that these size distributions are under
ambient conditions without an inlet drier. The ambient relative humidity for each air mass
back trajectory cluster was: Eastern Mediterranean (53%), Central Europe (61%), Atlantic
(74%), Western Mediterranean (70%), Mistral (low) (67%) and Mistral (high) (74%).
Consistent with the higher concentrations of sulphate and ammonium species, the "Eastern
Mediterranean" and "Central Europe" had the most pronounced accumulation modes with
respect to those from other clusters due to the presence of accumulation mode sulphate (see
Supplementary Figure S6 for the size-resolved chemical composition). In contrast, the
"Mistral (high)" air masses had very few particles in the accumulation mode and were mostly
dominated by nucleation and Aitken mode particles, in terms of number. There was only one
period of "Mistral (high)" air masses, spanning 38 hours between 09:00 on 24 June until 23:00
on 25 June.
The most pronounced new particle formation (NPF) events and subsequent growth were
observed during the two "Mistral" air masses, particularly between 25 and 27 June. Very high
number concentrations in the nucleation mode were also observed in very brief periods
during the Atlantic and, to an extent, the "Western Europe" air masses. There was a no trend
($R^2$ = 0.03) over the whole campaign between the "nucleation-mode ratio" (defined here as
the ratio of particles between 14 and 25 nm and 14 and 600 nm) and the fraction of $CH_3SO_2^+$
(a fragment of MSA, measured by the cToF-AMS) to total $PM_1$ organics (see Supplementary
Figure S7. There was a weak positive trend during periods of the Mistral (high) air mass ($R^2$ =
0.39). No trends were observed between the nucleation-mode ratio and the occurrence of



other cToF-AMS fragments such as amines that could be linked with biogenic gas-to-particle
conversion. Furthermore, there was a weak negative trend between the nucleation-mode
ratio and the calculated $f_{44}$ ($R^2 = 0.12$). Without instrumentation to measure the concentration
of clusters and smaller aerosols (<14 nm) in conjunction with organic vapours over a longer
time period, it is difficult to isolate and conclude the origin of these nucleation particles in a
general sense and we will limit our analysis to the most pronounced event during the
campaign. Figure 12 shows the size distribution over this period as well as $SO_2$, eBC and
$CH_3SO_2^+$, as well as the back trajectory ending at 04:00 UTC on 25 June at Lampedusa.
This NPF event occurred during the night and therefore in the absence of photochemistry.
There was no discernible increase in eBC or $SO_2$ during these events and the 3-day cascade
impactor sample from 25 until 28 June was characterized by the lowest concentrations of
vanadium and nickel (released from heavy oil combustion events due to ship emissions) of
the whole campaign.  The air mass backwards trajectory during this event was characteristic
of the "Mistral (high)" cluster. These are high altitude air masses descended over the Atlantic
Ocean before having undergone a hydraulic jump over the southern France region and then
a rapid descent over the western Mediterranean basin at high speed before arriving at the
Lampedusa site. It is interesting to note that these air masses were anomalous for the typical
June/July period at Lampedusa. Although the detected mass of $CH_3SO_2^+$ is likely due to the
condensation of MSA on accumulation mode particles, and considering that the cTof-AMS
collection efficiency below ~100 nm is poor, the increasing concentration of $CH_3SO_2^+$ did
coincide with the nucleation events during this period, suggesting a possible nucleation and
condensation of marine biogenic vapours. Different studies indicate that NPF events may be
triggered by atmospheric mixing processes (Kulmala et al., 2004; Hellmuth, 2006; Lauros et
al., 2007; Lauros et al., 2011) due to different phenomena like the enhancement of turbulence
in elevated layers (Wehner et al., 2010), the break-up of the nocturnal inversion (Stratmann
et al., 2003), or the turbulence associated with the nocturnal low-level jet (Siebert et al.,
2007). Furthermore, the intrusion of descending mid-tropospheric air masses in the boundary
layer (Pace et al., 2015) has been linked to the occurrence of NPF events and would be
consistent with the absence of high concentrations of BC, $SO_2$, vanadium and nickel which
could be expected from ship emissions in the boundary layer over the Mediterranean.



Pace et al. (2006) have shown that clean marine aerosol conditions are rare at Lampedusa
and generally associated with north-westerly progressively descending trajectories, in
agreement with the findings of this study. The relative absence of pre-existing particles acting
as a condensation sink favors NPF events as observed during the field campaign.

### 3.6 Evidence of aging across the Mediterranean

In order to investigate the aging of aerosols across the north-south trajectory of European
continental air masses, we compare the average NR-PM$_1$ composition measurements at
Lampedusa to the concurrent measurements conducted at the Ersa site during summer 2013
(Michoud et al., 2016; Arndt et al., 2017). This is shown in Table 2.
On average, the PM$_1$ non-refractory organic mass concentrations at both sites were similar
with ~3 µg m$^{-3}$. NO$_3^-$ concentrations were relatively small at both sites, but higher at Ersa (0.28
µg m$^{-3}$) than at Lampedusa (0.09 µg m$^{-3}$). Sulphate concentrations were a factor of 3.2 times
higher at Lampedusa (4.5 µg m$^{-3}$) than at Ersa (1.4 µg m$^{-3}$) while the ammonium
concentrations were a factor of 2.7.
To investigate the possible accumulation of ammonium sulphate during the transport of air
masses from Europe, the hourly air-mass back trajectories from Lampedusa were filtered so
that only those that passed within ±1° latitude and longitude and within ±200 m altitude of
the station height of Ersa (550 m) were selected. These thresholds were chosen arbitrarily
since there is no clear distinction in horizontal or vertical distance from the site that would
necessarily constitute a representative air mass. This resulted in a total of 192 hourly
observations at the Ersa site over 32 unique air mass backward trajectory runs (see Figure

541  13).

These trajectories were grouped mainly into the "Central Europe" cluster (n = 12). The median
trajectory duration between the Ersa and Lampedusa sites was 53 hours, with a minimum of
33 hours and a maximum of 144 hours (corresponding to the total duration of the HYSPLIT
model runs in this case). Those air masses grouped in the "Eastern Mediterranean" cluster
had the longest duration time between the sites of 127 hours (while coincident with the Ersa
site, these air masses still spent a significant amount of time over the eastern Mediterranean),
followed by "Central Europe" (83 hours), "Western Europe" (45 hours) and then the "Mistral
(low)" (38 hours). In general, between the two sites, there was a 40% enhancement in the



organic mass concentration, but an increase in sulphate and ammonium by a factor of 6 and 4, respectively (Table 2).

The accumulation of $(NH_4)_2SO_4$ between Ersa and Lampedusa appeared to be dependent on the travel time of the air mass, however different relationships were observed during different air mass clusters. The total sulphate concentration at Lampedusa minus the total sulphate concentration at Ersa for the same air mass and accounting for the travel time as a function of the travel time is shown in Figure 14.

There was a good positive correlation between the difference in sulphate concentrations between the two sites and the travel time for the "Central Europe" and "Eastern Mediterranean" air masses, while weak positive correlations were observed for the "Western Europe" and "Mistral (low)" clusters. It should be pointed out that the travel time was more than 110 hours for the "Eastern Mediterranean" air masses, while it was only between 33 and 58 hours for the other three air mass clusters. It is expected that the accumulation of sulphate would increase as the total travel time increases due to the opportunity for $SO_2$ conversion. Although this relationship is somewhat demonstrated here, there are other factors that would influence the $SO_4^{2+}$ accumulation. The sulphate concentrations presented here are measured by an ACSM and cToF-AMS at the Ersa and Lampedusa sites, respectively. Both of these instruments have a 100% inlet efficiency between ~100 nm and 800 nm. The conversion of $SO_2$ to $SO_4^{2-}$ via nucleation and condensation is dependent on the pre-existing aerosol size distribution and condensation sink. Therefore, the use of $PM_1$ composition can be misleading if the sulphate is condensing on coarse particles. This is demonstrated in Supplementary Figure S2 that shows the size-resolved mass distribution of sulphur and sodium collected every 3 days on multi-stage cascade impactor filters; the relative contribution of sulphur in the $PM_1$ is higher than that of $PM_{10}$ in the absence of sodium (a tracer for sea salt). Furthermore, the concentrations of $SO_4^{2-}$ measured by the PILS in the $PM_{10}$ fraction and cToF-AMS in the $PM_1$ are approximately equal with low sea salt concentrations (Na+ < 2 $\mu gm^{-3}$), but are nearly a factor of two higher with the PILS for higher sea salt concentrations (Supplementary Figure S3). Furthermore, the emission of $SO_2$, typically from ships in the Mediterranean, is not necessarily constant over time and is likely not uniformly spread over the basin and within the vertical column (e.g., Becagli et al. (2017)). This could possibly explain the discrepancy between the "growth rate" of sulphate between the Eastern Mediterranean and Central European air mass origins. Furthermore, the sample size for this analysis is



relatively small and potentially not representative of the general accumulation of $SO_4^{2-}$ but
nonetheless they highlight the magnitude of growth under different air mass trajectories.



## 4. Concluding remarks


The measurements carried out at Lampedusa during the ChArMEx/ADRIMED SOP-1a field
campaign has provided a unique insight into the surface layer aerosols in the remote Central
Mediterranean. Air masses were influenced by transport from the eastern Mediterranean,
central Europe, the western Europe, the Atlantic Ocean as well as western Europe. Air mass
clustering has been performed to explain observed differences in the aerosol composition
and size at Lampedusa.
Hourly $PM_1$ mass ranged from 1.9 to 33.4 μg m$^{-3}$, with an average of 10.2 μg m$^{-3}$. It was
composed on average of 41% ± 9% sulphate, 31% ± 8% organics, 17% ± 3% ammonium, 6% ±
4% black carbon, 1% ± 0.4% nitrate and 3% ± 2% sea salt. OA was highly oxidized ($f_{44}$ ~0.26),
and was apportioned to more oxidised oxygenated OA factor (MO-OOA, 53%), less oxidised
OOA factor (LO-OOA, 28%), methanesulfonic acid OOA (MSA-OOA, 12%) and to hydrogen-like
OA (HOA, 8%). The highest $PM_1$ mass loadings were observed for air masses from the Eastern
Mediterranean and central Europe, mostly due to the accumulation of ammonium and
sulphate. Ancillary data from a remote site at the northern point of Cape Corsica in the
Western Mediterranean showed increases of $SO_4^{2-}$ concentrations between 2 and 12 μg m$^{-3}$
when both sites (Corsica and Lampedusa) were connected. Apart from the dominance of
ammonium sulphate on the $PM_1$ composition, the mass concentration and sources of OA
have shown to be comparable to previous observations at European coastal and remote sites
in the Mediterranean. The most pristine air masses, in terms of $PM_1$, were observed during
periods with north-westerly winds which originated from the western Mediterranean or at
high altitudes over the western European continent. Several nucleation and growth events,
as well as large sea salt concentrations were observed during these pristine periods. The
largest concentrations of $PM_1$ were observed from air masses from central Europe and those
that had circulated over the eastern Mediterranean. In contrast to previous measurements
of column-integrated aerosol optical properties (Pace et al., 2006; Meloni et al., 2006), we did
not observe the presence of dust or biomass burning in the $PM_1$ range at the surface.



Our results also indicate a clear dichotomy of $PM_1$ aerosol composition from different source
regions. Air masses from central Europe were characterised by a higher organic fraction than
those from the eastern Mediterranean, which were enriched in sulphates. This difference
could have potential implications on the optical properties and particularly the cloud
condensation nuclei capabilities of those air masses. The relative occurrence of easterly air
masses is not evident in the climatological wind roses, nor in a previous study by Pace et al.
(2006) that took a climatological approach of the air mass back trajectories arriving at
Lampedusa from 2001 - 2003.  Nonetheless, a re-evaluation of the relative importance and
occurrence of different air masses and aerosol properties should be undertaken.

*Data availability*. Open-access to the data used for this publication is provided to registered
users    following    the    data    and    publication    policy    of    the    ChArMEx    program
(http://mistrals.sedoo.fr/ChArMEx/ Data-Policy/ChArMEx_DataPolicy.pdf). Additional code
used in the analysis of data can be obtained upon request from the corresponding or first
author. Weekly GDAS1-analysis (Global Data Assimilation System; 1° resolution) trajectory
files were downloaded from the Air Resources Laboratory (ARL) of the National Oceanic and
Atmospheric Administration (NOAA) archive (ftp://arlftp.arlhq.noaa.gov/archives/gdas1/).
144-hour air-mass backwards trajectories were calculated using the R-package, SplitR
(https://github.com/rich-iannone/SplitR). Cluster analyses were performed on these
calculated trajectories, using the R-package, OpenAir (Carslaw and Ropkins, 2012);
https://github.com/cran/openair). Spectra used for comparison of PMF OA factors from
those observed in other studies can be found at http://cires.colorado.edu/jimenez-
group/AMSsd/

*Author contributions.* PF, BD, KD, JFD, AGdS designed the experiment in Lampedusa, JS
designed the experiment in Ersa, with contributions of co-workers. BD, AM, MCB, FC, GP, KD,
CDB, JFD, MM, DM, JS, PZ, AGdS and PF performed the experiments. MDM, BD, AM, GP, KD,
JS and PF analysed data and all authors contributed to data interpretation. MDM, BD, GP, KD
and PF wrote the manuscript with contributions and/or comments from all co-authors.

*Competing interests.* The authors declare that they have no conflict of interest.




*Special issue statement.* This article is part of the special issue of the Chemistry and Aeosols

Mediterranean Experiment (ChArMEx) (ACP/AMT inter-journal SI)". It is not associated with

a conference.


*Acknowledgements*. This work is part of the ChArMEx project supported by CNRS-INSU,

ADEME, Météo-France and CEA in the framework of the multidisciplinary program MISTRALS

(Mediterranean Integrated Studies aT Regional And Local Scales; http://mistrals-home.org/).

It has also been supported by the French National Research Agency (ANR) through the

ADRIMED program (contract ANR-11-BS56-0006). Measurements at Lampedusa were also

supported by the Italian Ministry for University and Research through the NextData and

RITMARE Projects. The AERIS national data infrastructure is acknowledged for maintaining

the ChArMex database.






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





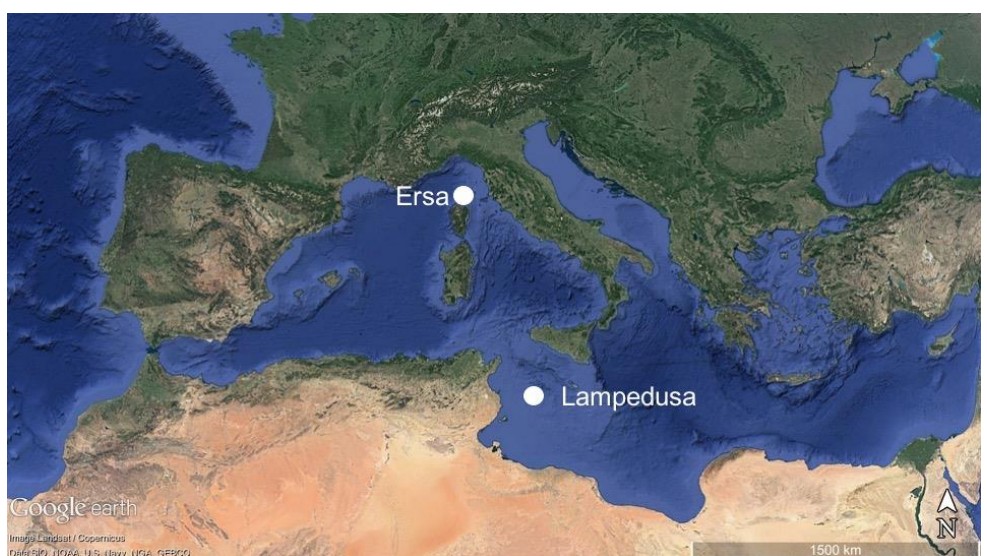


*Figure 1 The Mediterranean basin. The two sites considered in this study, Lampedusa and Ersa, are indicated with white*

*dots. Image is courtesy of Google Earth.*


*Figure 2 a) Hourly 144-hour (6 days) backwards trajectories from Lampedusa from 10 June 2013 until 5 July 2013, cut off at*
*120 hours (5 days). Colors represent the assigned cluster. B) The same as a) but cut off at 24-hour.*



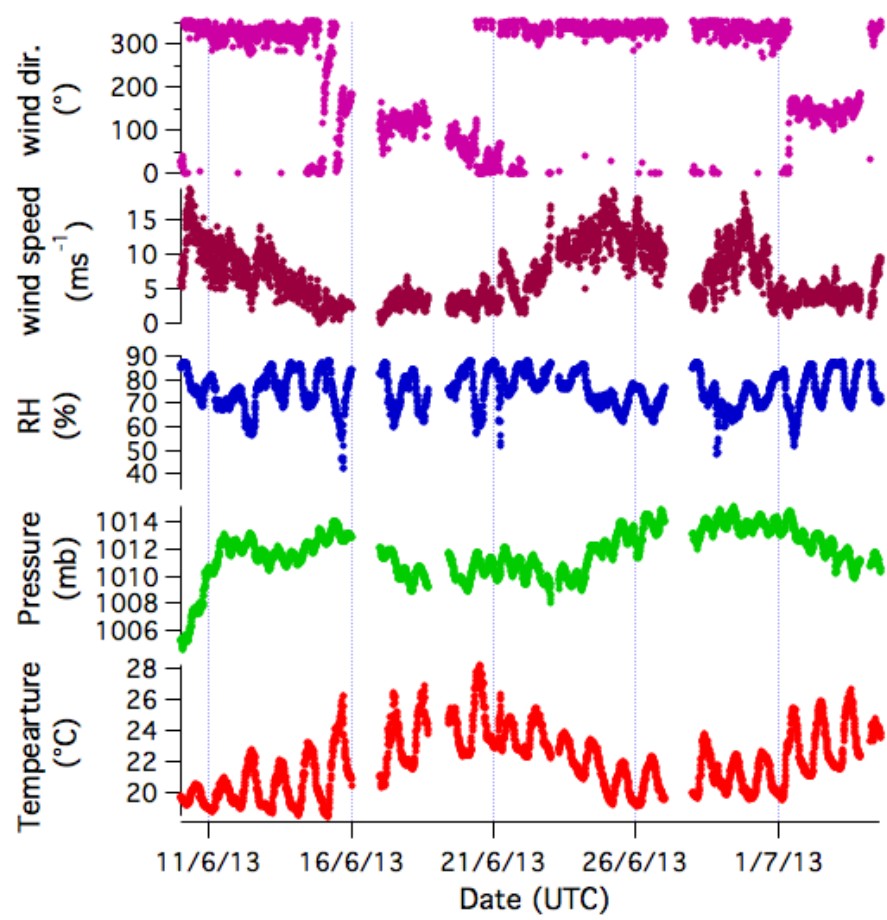



*Figure 3 Meteorological conditions (wind direction and speed, relative humidity, pressure and temperature) measured at the*
*Lampedusa site during SOP-1a.*




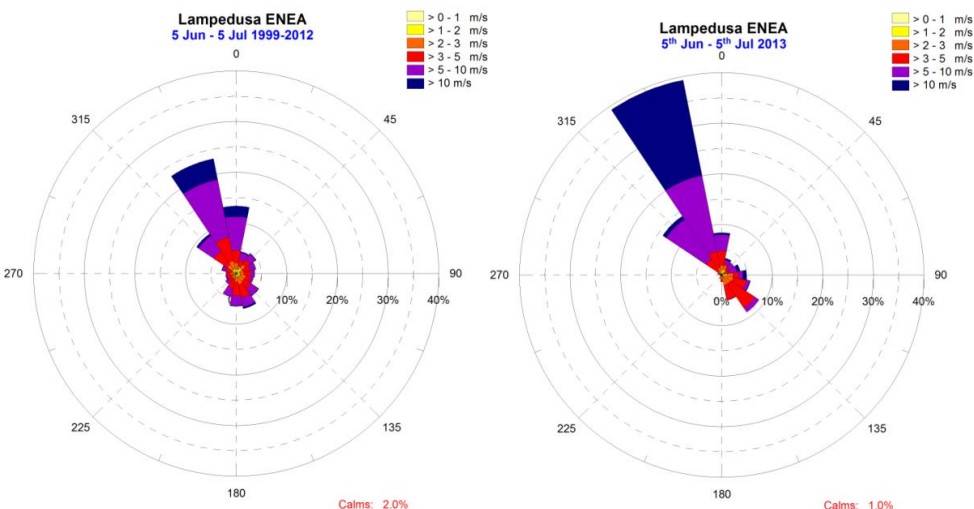

*Figure 4 Wind speed and direction at Lampedusa during the period from 5 June until 5 July during the years from 1999 - 2012*

*(left) and during this campaign (right).*

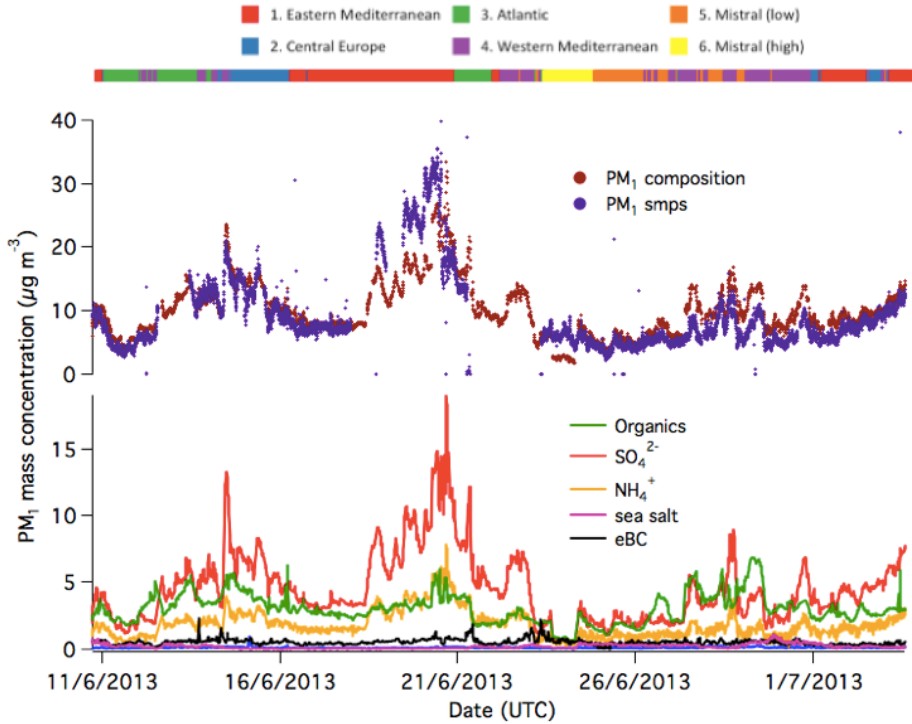

*Figure 5 The time series of PM$_1$ mass concentration, coloured by the relative contribution from each species. The top bar is*

*coloured according to the air mass origin.*



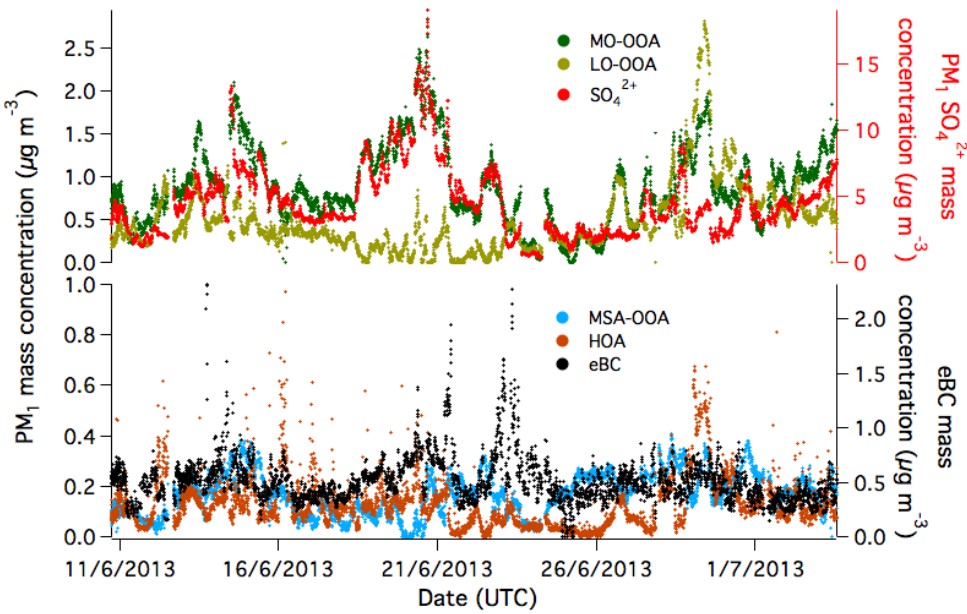


*Figure 6 The time series of the PM₁ "more oxidised" OOA (MO-OOA), "less oxidised" (OOA) and sulphate (top panel) and*

*methanesulfonic acid-related OOA (MSA-OOA), hydrocarbon-like OA (HOA) and eBC (bottom panel).*





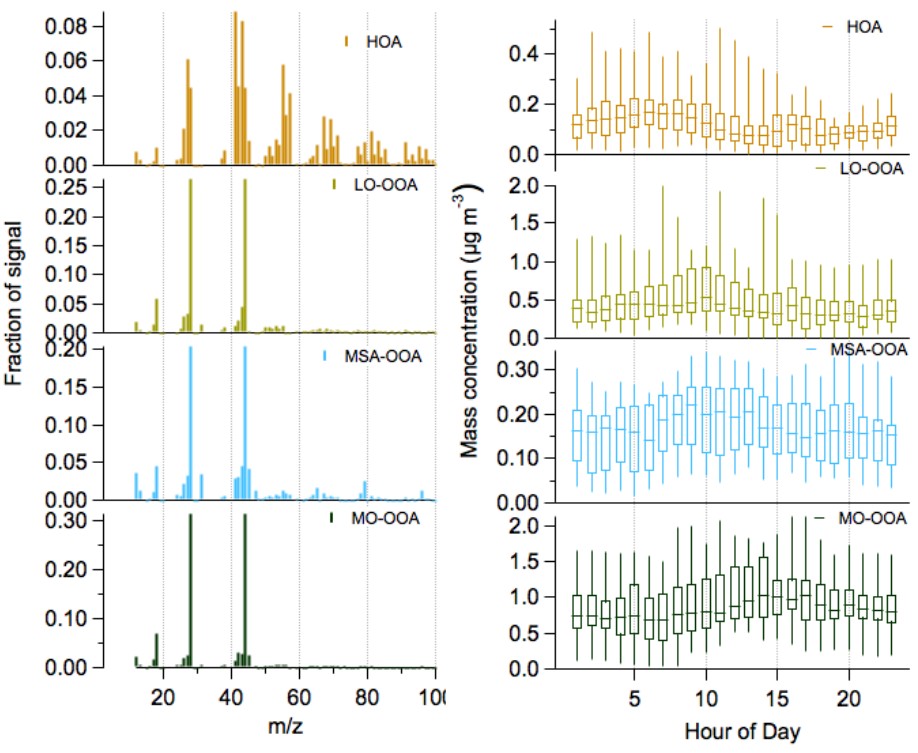


*Figure 7 The mass spectra for the 4 PMF factors (HOA: hydrocarbon-like organic aerosol, LO-OOA: less oxidised OOA, MO-*
*OOA: more oxidised OOA, MSA-OOA: methanesulfonic acid-related OOA) retrieved from the PMF analysis of unit-mass*
*resolution data.*
*Table 1 A summary of studies that have investigated NR-PM$_1$ composition (including PMF of OA) on islands within the*
*Mediterranean basin and at coastal sites surrounding the basin. Only studies that have investigated PMF-based OA source*
*apportionment                                    are                                    reported.*
*HOA: Hydrocarbon-like Organic Aerosol, SV-OOA: Semi-volatile oxygenated Organic Aerosol, LV-OOA: Low-volatility*
*oxygenated Organic Aerosol, BBOA: Biomass burning Organic Aerosol, COA: Cooking Organic Aerosol, OOA: Oxygenated*
*Organic Aerosol, F4: "Factor -4" (unidentified PMF factor), IndOA: Industry-related Organic Aerosol, OB-OA: "Olive-branch*
*Organic Aerosol. PMF factors in bold indicate secondary organic aerosol. After the results of this study, observations are*
*ordered according to longitude (west to east).*

| AUTHORS (YEAR) | LOCATION | PERIOD | INSTRUMENT | PM$_1$ MASS AND COMPOSITION | PMF FACTORS |
|---|---|---|---|---|---|
| **THIS STUDY** | Lampedusa (35°31'5"N, 12°37'51"E, 45 m a.s.l.) | 10 June - 5 July 2013 (summer) | cToF-AMS | 10.1 µg m$^{-3}$ (OA: 30%, SO$_4^{2-}$: 44%, NH$_4^+$: 18%, NO$_3^-$: 1%, seasalt: 1%, eBC: 5%) | HOA (8%)<br>**MSA-OOA (12%)**<br>**LO-OOA (28%)**<br>**MO-OOA (53%)** |


| Reference | Location | Period | Instrument | Composition | OA factors |
|---|---|---|---|---|---|
| (MINGUILLÓN ET AL., 2015) | Montseny (41°46'46"N, 02°21'19"E, 720 m a.s.l.) | June 2012 - July 2013 | ACSM | Summer: 10.8 µg m⁻³ (OA: 60%, SO₄²⁻: 20%, NH₄⁺: %, NO₃⁻: %, Winter: 6.3 µg m⁻³ (OA:, SO₄²⁻: 8%, NH₄⁺: %, NO₃⁻: %, | Summer: HOA (13%) **SV-OOA (41%)** **LV-OOA (44%)** Winter: HOA (12%) BBOA (28%) **OOA (59%)** |
| (ARNDT ET AL., 2017) | Cape Corse (42°58'8.4"N, 9°22'48"E, 544 m a.s.l.) | 11 June - 6 August 2013 (Summer) | Q-ACSM | 5.5 µg m⁻³ (OA: 55%, SO₄²⁻: 26, NH₄⁺: 13%, NO₃⁻: 5%) | **SV-OOA (62%)** **LV-OOA (33%)** |
| (MICHOUD ET AL., 2017) | Cape Corse (42°58'8.4"N, 9°22'48"E, 544 m a.s.l.) | July 15 - August 5 2013 (Summer) | Q-ACSM | 6.8 µg m⁻³ (OA: 55%, SO₄²⁻: 27%, NH₄⁺: 13%, NO₃⁻: 5%) | HOA (4%) **SV-OOA (44%)** **LV-OOA(53%)** |
| (RINALDI ET AL., 2017) | Cape Granitola (37°34'31.1"N, 12°39'34.2"E, 5 m a.s.l.) | April 2016 (Spring) | HR-ToF-AMS | 3.5 µg m⁻³ (OA: 37%, SO₄²⁻: 31%, NH₄⁺: 12%, NO₃⁻: 3%, seasalt: 10%, BC: 6%) | HOA (3%) BBOA (2%) **OOA-1 + OOA-2 (70%)** **OOA-3 (25%)** |
| HILDEBREANDT ET AL., 2010 | Finokalia (35°20'N, 25°40'E, 150 m a.s.l.) | May 2008 (Spring) | Q-AMS | 9 µg m⁻³ (OA: 28%, SO₄²⁻: 55%, NH₄⁺: 16%, NO₃⁻: 2%) | OOA-1 (61%) OOA-2 (39%) |
| HILDEBRAND ET AL., 2011 | Finokalia (35°20'N, 25°40'E, 150 m a.s.l.) | 25 February - 26 March 2009 (late Winter) | Q-AMS | 3.3 µg m⁻³ (OA: 43%, SO₄²⁻: 42%, NH₄⁺: 14%, NO₃⁻: 2%) | **OOA (>56%)** OB-OA (15 - 35%) Amine-OA (6 - 21%) |
| BOUGIATIOTI ET AL., 2014 | Finokalia (35°20'N, 25°40'E, 150 m a.s.l.) | August - September 2012 | Q-ACSM | Fire events: OA: 46.5% , SO₄²⁻: 29.2%, NH₄⁺: 11.7%, NO₃⁻: 3.2%, BC: 9.5% Nonfire periods: OA: 34.7% , SO₄²⁻: 43%, NH₄⁺: 13.7%, NO₃⁻: 2.2%, BC: 6.1% | BBOA (<20%) **OOA1-BB + OOA2 (>80%)** |






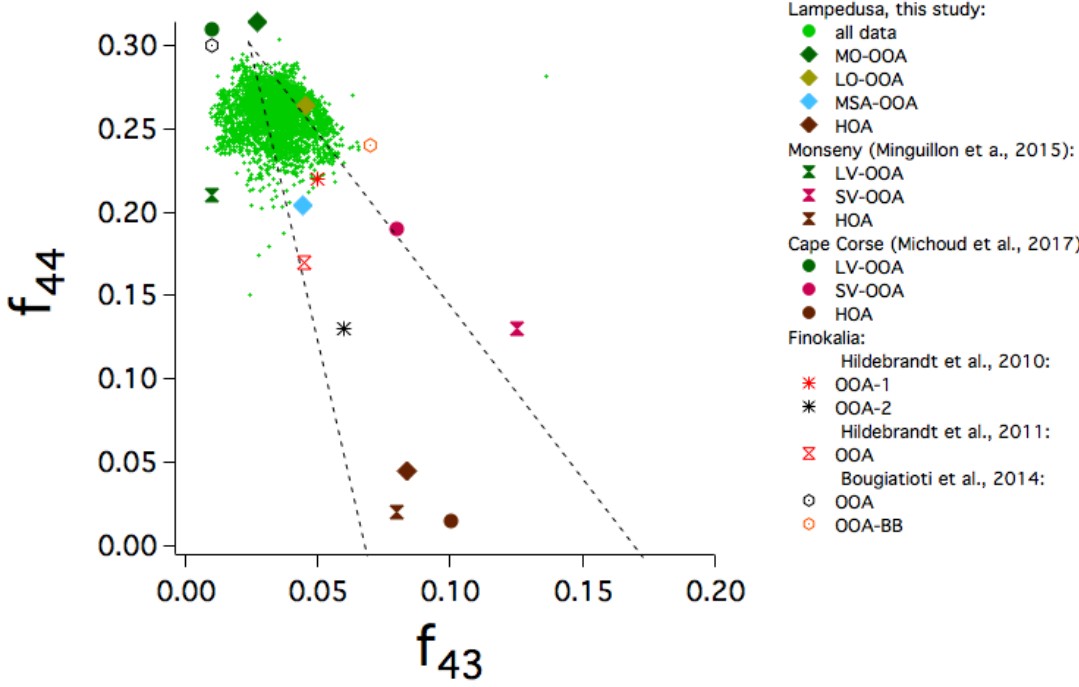


*Figure 8 $f_{43}$ (the ratio of m/z 43 to the total OA) against $f_{44}$ (ratio of m/z 44 to the total OA). The triangle is considered to*
*encapsulate typical atmospheric values of OA according to Ng et al. (2010). The values for the various PMF factors from this*
*study and other studies conducted in the remote Mediterranean are also displayed.*





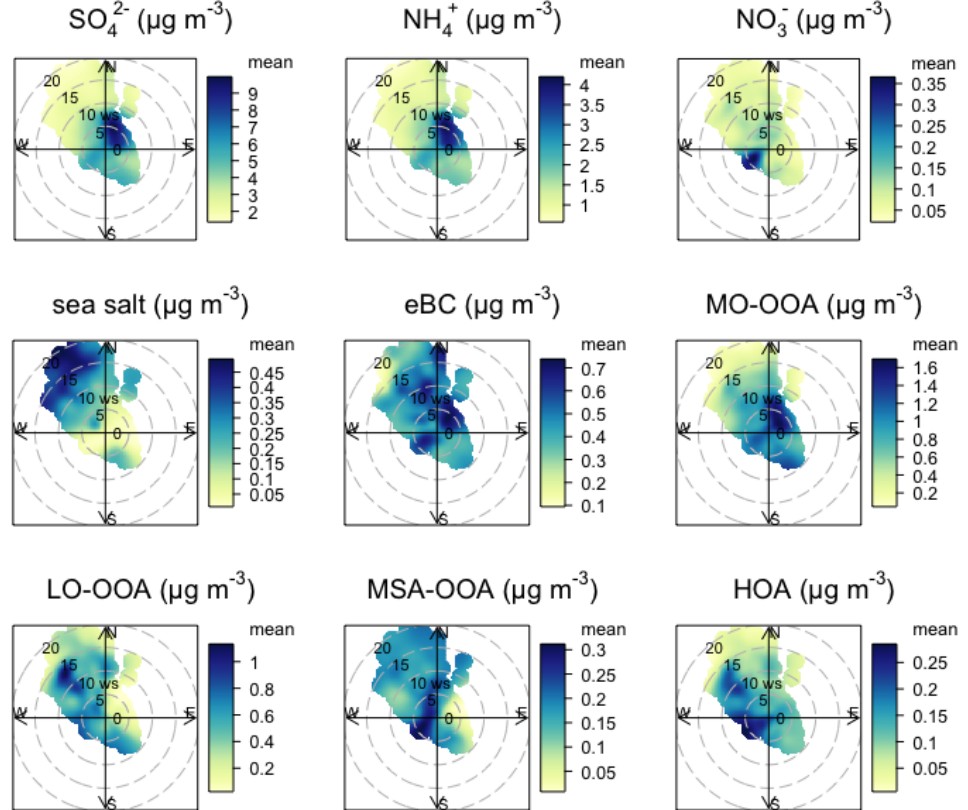


*Figure 9 Bi-variate polar plots of mean concentrations of PM$_1$ species and f$_{44}$ at Lampedusa. The angle represents the arrival*
*wind direction, the radius represents the wind speed and the colours represent the mean concentrations for the respective*
*wind directions and winds.*



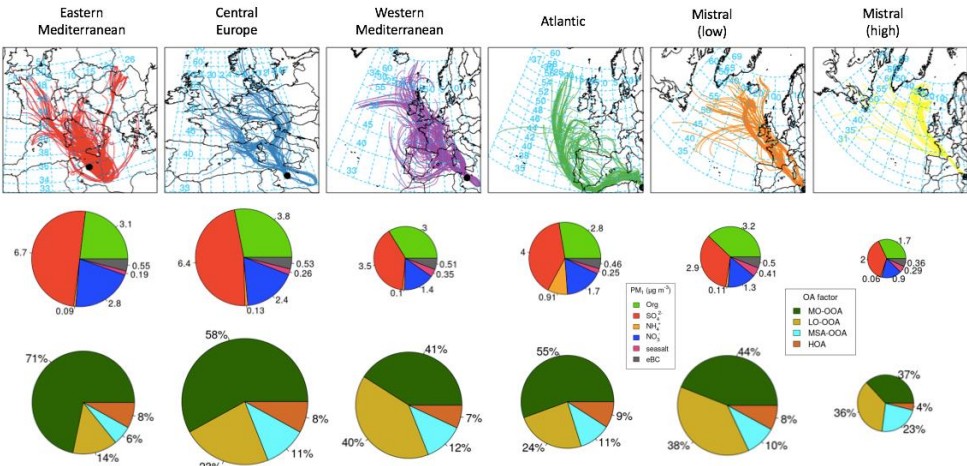

*Figure 10 top) 144 hour air mass back trajectories, assigned to each cluster; middle) the PM$_1$ composition for each air mass*

*cluster and bottom) the contribution of OA factors for each air mass cluster. The diameter for the PM$_1$ composition pie graphs*

*are proportional to the total PM$_1$ concentration for each air mass cluster period and the radius for the OA factor pie graphs*

*is proportional to the total PM$_1$ organic concentration for each air mass cluster period.*

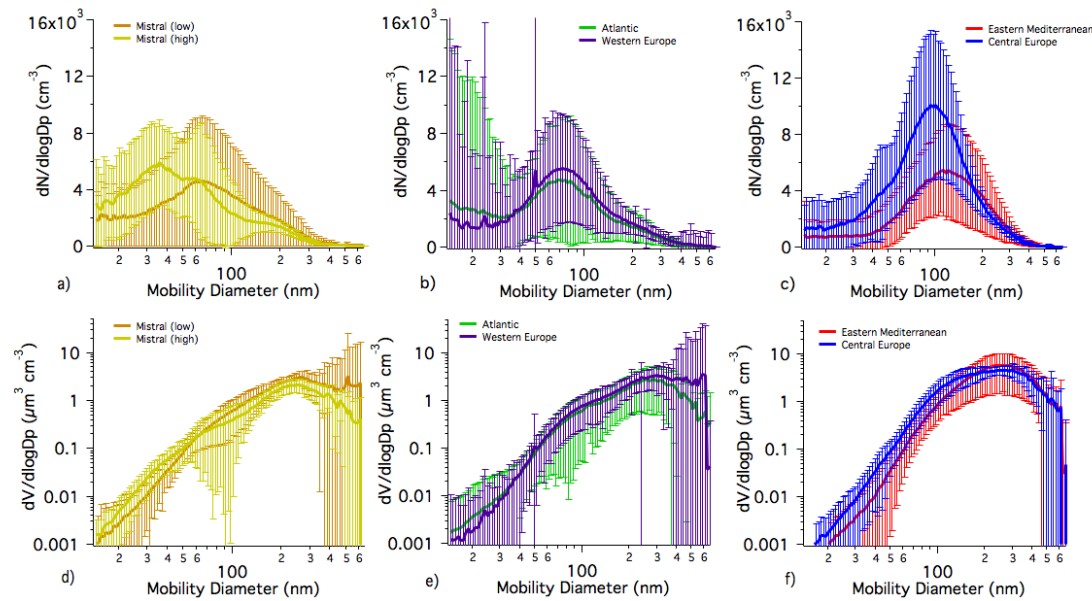

*Figure 11 The number size distribution of PM$_1$ aerosol, coloured by averages for different air mass origin: a) Mistral (high)*

*and Mistral (low); b) Atlantic and Western Europe; c) Eastern Mediterranean and Central Europe and the volume size*

*distribution of PM$_1$ aerosol, coloured by averages for different air mass origin: d) Mistral (high) and Mistral (low); e) Atlantic*

*and Western Europe; f) Eastern Mediterranean and Central Europe.*



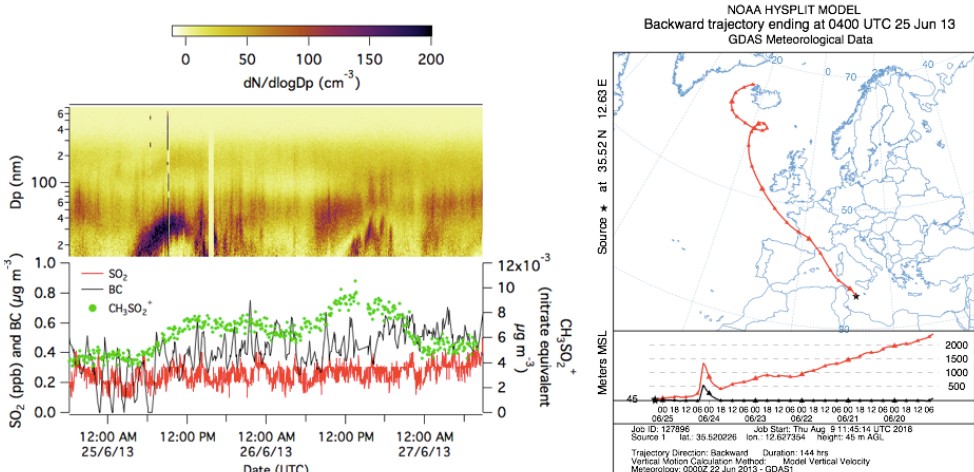

979

Figure 12 (left) The number size distribution during a new particle formation event on 25 June 2013 and the corresponding

concentrations of $SO_2$, eBC and MSA fragment, $CH_3SO_2^+$, and (right) the HYSPLIT air mass backwards trajectory during the

event.

Table 2 Campaign average $PM_1$ concentration for the major aerosol species measured at the Ersa and Lampedusa sites during

the SOP-1a period and for periods of coincident air mass backwards trajectories between Ersa and Lampedusa

| SITE | $SO_4^{2-}$ | ORGANIC | $NH_4^+$ | $NO_3^-$ |
|---|---|---|---|---|
| ERSA | 1.4 ± 2.6 | 3.0 ± 1.1 | 0.7 ± 1 | 0.3 ± 0.1 |
| LAMPEDUSA | 4.5 ± 0.9 | 3.0 ± 1.6 | 1.9 ± 0.5 | 0.1 ± 0.2 |
| ERSA (COINCIDENT WITH LAMPEDUSA) | 0.9 ± 0.5 | 2.7 ± 1.1 | 0.5 ± 0.3 | 0.4 ± 0.3 |
| LAMPEDUSA (COINCIDENT WITH ERSA) | 5.3 ± 2.0 | 3.8 ± 0.8 | 2.0 ± 0.6 | 0.1 ± 0.1 |




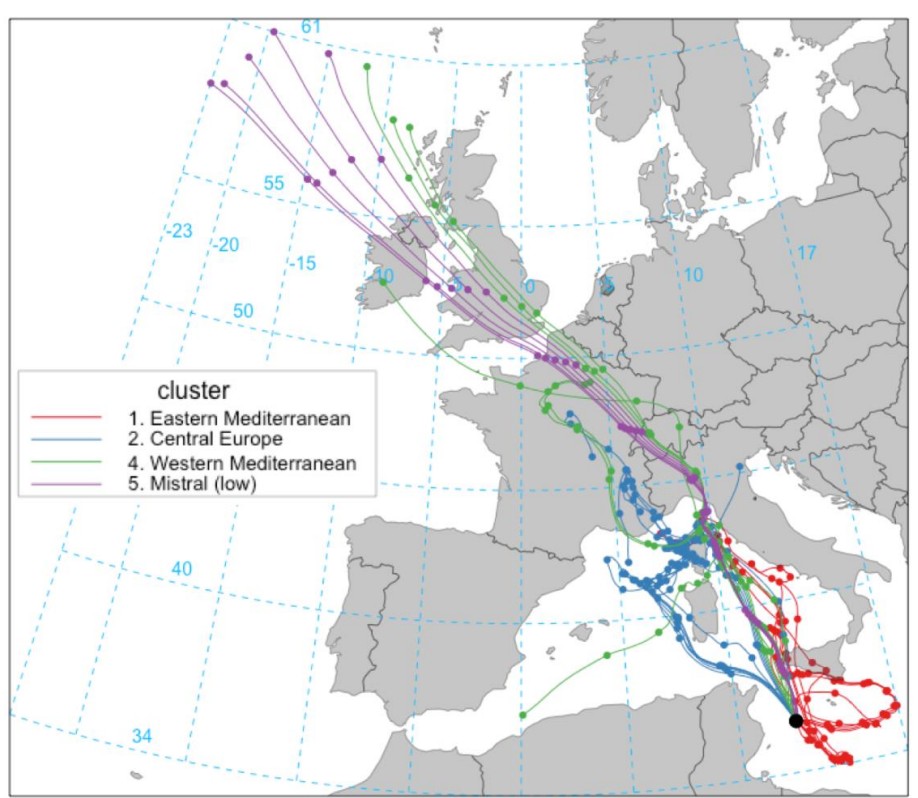


*Figure 13 Hourly 120-hour (5 days) backwards trajectories from Lampedusa that passed within ±1° in latitude and longitude*

*and ±200 m in altitude of the Ersa station. The colours represent the assigned cluster (performed on 144 hour trajectories).*




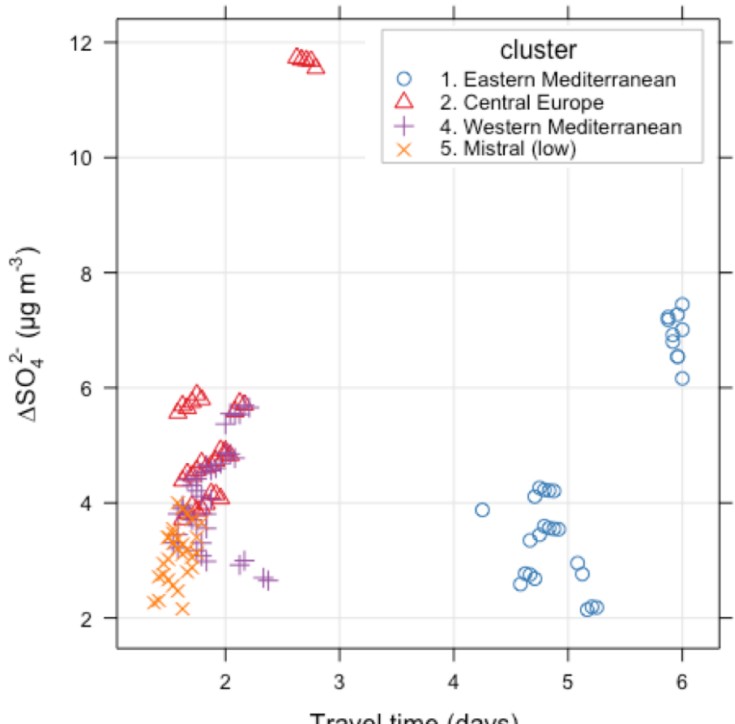


*Figure 14 The difference in the PM$_1$ SO$_4^{2-}$ mass concentration at Lampedusa and Ersa as a function of the travel time of the*
*air masses from Ersa to Lampedusa. Colours represent the air mass origin cluster.*