# Peer review of "Summertime surface PM1 aerosol composition and size by source region at the Lampedusa island in the central Mediterranean Sea"

_Atmospheric Chemistry and Physics, 2019_

## Referee Comment (RC1) · Anonymous Referee #1 · 28 Mar 2019

The manuscript of Mallet et al. presents chemical composition and size distribution measurements conducted at the island of Lampedusa, Italy during a one-month period in the summer of 2013. It occurs that ammonium sulfate is the main contributor (63%) to the submicron non-refractory mass, followed by organics (33%). By performing Positive Matrix Factorization (PMF) analysis on the derived organic aerosol mass spectra it occurs that there are four factors contributing to the total organic aerosol, namely a hydrocarbon-like OA, a methanesulfonic acid-related OA, and two oxidized OA, a more-oxidized and a less-oxidized one. The two secondary OOA factors contribute the most (more than 80%) to the total OA, but with having different origin. The more-oxidized was observed during easterly air masses from the eastern Mediterranean and central

Europe while the less-oxidized during westerly winds from the western Mediterranean, the Atlantic Ocean and high altitudes over France and Spain from mistral winds. Finally, an attempt is made to investigate the aging of aerosols by comparing concurrent measurements at Lampedusa and Corsica, revealing a dependence on travel time between the two sites and an enhancement of organics (40%) and a significant increase in sulfate and ammonium (by a factor of 6 and 4, respectively) between Ersa (Corsica) and Lampedusa.

The paper is well written and easy to follow, though there are some issues and more thorough discussion should be made in specific sections. A very interesting point of the study is the study of the aging aerosol gradient and its dependence on the time travel of the air masses between Corsica and Lampedusa. Other than that the paper can be recommended for publication after addressing the issues listed below.

Specific comments:

1) More information about the c-ToF-AMS measurements and data analysis should be provided: - Response Factors and/or Relative Ionization Efficiencies of the different species

- Was there a collection efficiency correction applied?? Was a constant CE used or a chemical composition dependent one e.g. Middlebrook et al. (2012)?

Also I would suggest creating a separate section after Section 2.2 as Quality control/Quality assurance of the measurements where I would include the comparisons between PM1 from chemical composition and SMPS, sulfate from c-ToF-AMS and PILS and the supporting measurements from the nanoMOUDI.

2) On multiple occasions in the manuscript the term "agreement" is mentioned, but no actual metric is provided. For example, in L307 "reasonable agreement between the PM1 concentration calculated from composition measurements and the SMPS" is stated, but what does this translate to? Apart from the timeseries, no scatter plot is

provided, no correlation coefficient, therefore how is this agreement defined? Same in L312.

Technical corrections:

L76 I would also add here the references of Bougiatioti et al. (2014) and Minguillon et al. (2015) as identifying biomass burning aerosol in the Mediterranean during summer

L126 change to "BBOA"

L140 secondary sites established (delete "were")

L162 probably you mean Total Suspended Particulate (TSP)

L165 check font style

L157-164 More information on the c-ToF-AMS measurements should be provided here

L299 Dry NR-PM1? There is nothing mentioned about using a dryer in the instrumentation section (2.2)

L307 Reasonable agreement meaning what? R2 of how much?

L312-314 Do you mean between c-ToF-AMS and PILS? If yes I would suggest to change and state the methods used, preferably also give a correlation coefficient

L340 low-volatility/ highly oxidized

L340 Make title bold

L511-516 Night-time nucleation events have also been observed in the Eastern Mediterranean (Kalivitis et al. 2012)

Reference

Kalivitis, N., Stavroulas, I., Bougiatioti, A., Kouvarakis, G., Gagné, S., Manninen, H. E., Kulmala, M., and Mihalopoulos, N.: Night-time enhanced atmospheric ion concentrations in the marine boundary layer, Atmos. Chem. Phys., 12, 3627-3638,

https://doi.org/10.5194/acp-12-3627-2012, 2012.

---

## Referee Comment (RC2) · Anonymous Referee #2 · 12 Apr 2019

This study presents results on composition of fine PM fraction (approx. PM1) in Lampedusa, an island site in the southern central Mediterranean), by using a cToF-AMS. Results were obtained during the first CHARMEX Special Observation Period (SOP1) in summer 2013. Results were compared with similar studies performed in the Mediterranean region in different periods and specifically with those obtained at Ersa site, Corsica, during the sampling period. The novelty of this work lies in the fact that it is the first study of this type carried out on an island in the central Mediterranean. One of the main concerns of this study is the short duration of the sampling period (less than one month). This can affect its representativeness, the comparison with other studies and the interpretation of the results. However, results can be considered of interest in

the framework of the CHARMEX SOP1.

Variations in PM1 composition are interpreted as a function of the origin of air masses. Higher concentrations of sulfate were obtained during transport from eastern Mediterranean, probably due to the impact of emissions from this region. A clear variation was also observed for the LO-OOA/MO-OOA ratios, with a higher contribution of the most oxidized aerosols with transport form the east.

Authors attempt to study the aging of aerosols during transport by comparing PM1 composition at Lampedusa and Ersa, when affected by the "same" air masses. Comparison was performed for the different clusters defined. This comparison was mainly focused on sulfate; differences were related to the accumulation of SO4 and the SO2 conversion (mainly related to the shipping emissions). A significant increase was obtained for sulfate concentrations during transport of air masses form the East. As shown in Figure 13, during transport from eastern Mediterranean, it seems that the Lampedusa site is affected by other air masses different to those impacting at Ersa. Thus, higher concentrations of sulfate at Lampedusa may be related to the impact of air masses from the East, that are not impacting at Ersa. Therefore, the proposed methodology has some limitations for estimating the aging under these scenarios.

Minor changes

Line 99 (e.g. FLEXPART; (Stohl et al., 2005)

Line 108: (PMF; (Paatero, 1997;

Lines 144-146: is the first "detailed characterization" in the Central Mediterranean, at Lampedusa, or during the CHARMEX project?

Line 162: Please, indicate the sampling flow

Line 167: for major inorganic and organic. . .

Line 175: please indicate flow for MOUDI; did you use the same TSP inlet for all the

instructions?

Line 177: which kind of filters did you use?

Line 179: Denjean et al. (2016)).

Line 214: by (Ovadnevaite et al., (2012).

Line 215: High uncertainty estimation of the sea salt

Line 299: please, specify the time period of the concentrations (hourly basis; 30 minute?)

Lines 312, 313, 314, 565: $SO_4^{2-}$

Line 314: (see Supplementary Figure S2).

Lines 319-322: this is estimation, more measurements, for a wider period are necessary for demonstrating this.

Lines 416-418: This comparison will depend on the sampling periods. This study was performed in summer, where high concentrations of sulfate are expected. The study by El Haddad et al (2013), also in summer, showed higher concentration of sulfate than OA

Line 438. In figure 8: concentration of sulfate and ammonium seem higher during E – NE air masses; not north west as stated here; a similar pattern to that described for MO-OOA (Line 448).

Line 450: Pattern of LO-OOA is similar to that of HOA

Lines 463-466; Section 3.4. Figure 6. There is a clear difference between the ratio LO-OOA/MO-OOA during the eastern and western air masses; any comment on this? –

Section 3.5. The measurements of size distribution are limited to the 14-600 nm fractions. The lower size is relatively high for studying the nucleation episodes. Moreover

the different air humidity measured for the air clusters defined (sampling was at ambient conditions – Line 469-472) may affect these measurements.

Lines 485-487: Has the "nucleation mode ratio" been previously defined? Can you add a reference?

Line 501: Please, add a reference for shipping emissions

Figure 6. caption: "less oxidized" (LO-OOA) . . .

––––––––––––––––––––––––––––––––––

---

## Author Comment (AC1) · 3 Jul 2019

acp-2019-192: Summertime surface PM1 aerosol composition and size by source region at the Lampedusa island in the central Mediterranean Sea, Mallet et al., 2019

Author response to reviewers (RC1, RC2), written by Marc D. Mallet on behalf of all authors. Reviewer comments are indicated in bold, author comments are indicated in normal text and sections taken from the manuscript are indicated in italics.

**RC1.**

**The manuscript of Mallet et al. presents chemical composition and size distribution measurements conducted at the island of Lampedusa, Italy during a one-month period in the summer of 2013. It occurs that ammonium sulfate is the main contributor (63%) to the submicron non-refractory mass, followed by organics (33%). By performing Positive Matrix Factorization (PMF) analysis on the derived organic aerosol mass spectra it occurs that there are four factors contributing to the total organic aerosol, namely a hydrocarbon-like OA, a methanesulfonic acid-related OA, and two oxidized OA, a more oxidized and a less-oxidized one. The two secondary OOA factors contribute the most (more than 80%) to the total OA, but with having different origin. The more-oxidized was observed during easterly air masses from the eastern Mediterranean and central Europe while the less-oxidized during westerly winds from the western Mediterranean, the Atlantic Ocean and high altitudes over France and Spain from mistral winds. Finally, an attempt is made to investigate the aging of aerosols by comparing concurrent measurements at Lampedusa and Corsica, revealing a dependence on travel time between the two sites and an enhancement of organics (40%) and a significant increase in sulfate and ammonium (by a factor of 6 and 4, respectively) between Ersa (Corsica) and Lampedusa.**

**The paper is well written and easy to follow, though there are some issues and more thorough discussion should be made in specific sections. A very interesting point of the study is the study of the aging aerosol gradient and its dependence on the time travel of the air masses between Corsica and Lampedusa. Other than that the paper can be recommended for publication after addressing the issues listed below.**

Authors: The authors would like to thank the reviewer for her/his time and helpful comments. The reviewer's main concerns were surrounding the lack of reported details surrounding the c-ToF-AMS measurements and data analysis and comparisons made between the cToF-AMS, SMPS and PILS measurements. We address these specific comments in the following section, "Specific comments:".

In addition to the changes outlined below, several other changes in the manuscript have been made (e.g. reference formatting, typos). The major change is that data presented in Table 1 and Supplementary Table S1 has now been combined and displayed in a new figure (Figure 8; see below). This figure summarises the $PM_1$ composition from all of the previous studies around the Mediterranean basin that have performed a PMF analysis. This new Figure makes it much easier to compare the results of our study to these previous studies by visualising the composition with pie charts pointing to the sampling location, rather than presenting the data in tables. Subsequent Figure and table numbers have been updated.

[Figure]

*Figure 8 A summary of studies that have investigated NR-PM₁ composition (including PMF of OA) around the Mediterranean basin. Only studies that have investigated PMF-based OA source apportionment are reported. Pie charts display the average concentration during each study where green corresponds to organics, red to sulphates, orange to ammonium, blue to nitrate, pink to either chlorides or sea salt and black to elemental or black carbon. The OA fraction acronyms correspond to the following:HOA: Hydrocarbon-like Organic Aerosol, SV-OOA: Semi-volatile oxygenated Organic Aerosol, LV-OOA: Low-volatility oxygenated Organic Aerosol, BBOA: Biomass burning Organic Aerosol, COA: Cooking Organic Aerosol, OOA: Oxygenated Organic Aerosol, F4: "Factor -4" (unidentified PMF factor), IndOA: Industry-related Organic Aerosol, OB-OA: "Olive-branch Organic Aerosol. See Supplementary Tables S1 and S2 for further details about the sampling locations, instruments used and pie chart values. ¹This study collected on PM2.5 filters and nebulised into an HR-ToF-AMS. ²Excludes fire-periods.*

**Specific comments:**

**1) More information about the c-ToF-AMS measurements and data analysis should be provided: - Response Factors and/or Relative Ionization Efficiencies of the different species. Was there a collection efficiency correction applied?? Was a constant CE used or a chemical composition dependent one e.g. Middlebrook et al. (2012)?**

**Also I would suggest creating a separate section after Section 2.2 as Quality control/Quality assurance of the measurements where I would include the comparisons between PM1 from chemical composition and SMPS, sulfate from c-ToF-AMS and PILS and the supporting measurements from the nanoMOUDI.**

Authors: More detail has been given for the c-ToF-AMS measurements and data analysis in both sections 2.2 Instrumentation, measurements and data, and 2.3.1 Analysis of the cToF-AMS data. We did not include a separate QA/QC section but instead provided more detailed comments in the Experimental and Results section regarding the cToF-AMS measurements and comparisons between the SMPS and PILS.

With regards to the c-ToF-AMS ionization calibrations, relative ionization efficiencies and collection efficiencies, the following paragraph as been included in Section 2.2:

*"The ionization efficiency (IE) with respect to nitrate anions was calculated every 5-6 days using nebulised 350 nm mobility diameter ammonium nitrate particles (values varied between $1.42 * 10^{-7}$ and $1.53 * 10^{-7}$). The relative IE (RIE) of ammonium was slightly higher than the default value and was 4.3 based on the mass spectrum of ammonium nitrate data from IE calibrations. The RIE of sulfate was determined by comparing the theoretical and the measured concentration of a solution of ammonium nitrate and ammonium sulfate and was determined to be the default value of 1.2. For the organic fraction, the default value of 1.4 was used. For each of the major species, a composition dependent collection efficiency was applied as proposed by Middlebrook et al., 2012 and was on average 0.549, very similar to the default value of 0.55."*

Further detail of the sea salt estimation has also been provided:

*"The PM1 sea salt concentration was estimated in the cTof-AMS by applying a scaling factor of 102 to the ion fragment (using the cumulative peak fitting analysis described in Muller et al., 2011) at 57.98 assigned to NaCl as proposed by Ovadnevaite et al., 2012. This scaling factor was determined by nebulising monodisperse 300 nm (mobility diameter) NaCl particles into the cToF-AMS and comparing the $NaCl^+$ signal to the total mass calculated using the number concentration from a CPC-3010. This calibration was done after the campaign but with similar tuning conditions. The sea salt-$SO_4^{2-}$ ($ss$-$SO_4^{2-}$) was calculated as $0.252 * 0.3 * [seasalt]$, where 0.252 is the mass ratio of $SO_4^{2-}$ to $Na^+$ in sea salt and 0.3 is the mass ratio of $Na^+$ to sea salt (Ghahremaninezhad et al., 2016). Given these assumptions, the uncertainty in the seasalt concentrations are likely to be significantly higher than the typical 20%, although the total contribution of seasalt to the PM1 fraction was very small (0.30 µg $m^{-3}$; <4 %)."*

**2) On multiple occasions in the manuscript the term "agreement" is mentioned, but no actual metric is provided. For example, in L307 "reasonable agreement between the PM1 concentration calculated from composition measurements and the SMPS" is stated, but what does this translate to? Apart from the timeseries, no scatter plot is provided, no correlation coefficient, therefore how is this agreement defined? Same in L312.**

Authors: The reviewer is correct that "agreement" is too loose of a term. The first instance when it is used:

*"reasonable agreement between the PM1 concentration calculated from composition measurements and the SMPS..."*

Now reads as:

*"There was reasonable agreement (slope = 0.62; $R^2$ = 0.67) between the PM1 mass concentration calculated from composition measurements and the SMPS…"*

The second instance of when it was used:

*"For most of the campaign there was a good agreement between the $PM_1$ $SO_4$2+ and the TSP $SO_4$2+ concentration, with the exception of periods of high sea salt concentrations when the TSP SO 42+ were significantly higher (see Supplementary Figure S2)."*

Now reads:

*"During most of the campaign there was a reasonable agreement (slope = 1; $R^2$ = 0.6) between the PM1 $SO_4^{2-}$ (c-ToF-AMS) and the TSP $SO_4^{2-}$ (PILS) concentration, with the exception of periods of high sea salt concentrations when the TSP $SO_4^{2-}$ were significantly higher (slope = 0.5; $R^2$ = 0.2 for TSP Cl- concentrations > 10 µg $m^{-3}$; see Supplementary Figure S2)."*

**Technical corrections:**
**L76 I would also add here the references of Bougiatioti et al. (2014) and Minguillon et al. (2015) as identifying biomass burning aerosol in the Mediterranean during summer**

Authors: These references have been added. The sentence now reads :

*"Furthermore, biomass burning aerosol has frequently been observed over the basin, in particular the dry season in summer when forest fires are more common (Bougiatioti et al., 2014; Minguillon et al., 2015; Pace et al., 2005)."*

**L126 change to "BBOA"**

Authors:  The incorrectly labelled "BBA" has been changed to "BBOA".

**L140 secondary sites established (delete "were")**

Authors: The incorrect use of "were" has been deleted. The sentence now reads:

*"Numerous secondary sites established along the Mediterranean coasts in Spain, Italy and Corsica beyond the SOP-1a have also provided valuable knowledge of the atmospheric composition in the western and central Mediterranean regions (Chrit et al., 2017; Chrit et al., 2018; Becagli et al., 2017)."*

**L162 probably you mean Total Suspended Particulate (TSP)**

Authors: "Total Suspected Particulate (TPS)" has been changed to "Total Suspended Particulate (TSP)".

**L165 check font style**

Authors:  All fonts throughout the manuscript have now been made consistent.

**L157-164 More information on the c-ToF-AMS measurements should be provided here**

Authors: As detailed above, more details about the c-ToF-AMS measurements have been provided (IE, RIE, CE, seasalt scaling).

**L299 Dry NR-PM1? There is nothing mentioned about using a dryer in the instrumentation section (2.2)**

Authors: A nafion drier was used. The relative humidity at the AMS inlet was logged for ~ half of the campaign and was below 55%. This has been indicated in Section 2.2:

*"A nafion drier was used, however the relative humidity at the inlet of the c-ToF-AMS was checked throughout the campaign and was always below 55%."*

**L307 Reasonable agreement meaning what? R2 of how much?**

Authors: As described above, a slope and correlation coefficient for the PM1 from the cToF-AMS and SMPS has now been provided:

*"There was reasonable agreement (slope = 0.62; $R^2$ = 0.67) between the PM1 mass concentration calculated from composition measurements and the SMPS..."*

**L312-314 Do you mean between c-ToF-AMS and PILS? If yes I would suggest to change and state the methods used, preferably also give a correlation coefficient**

Authors: As described above, a slope and correlation coefficient has been provided. We have also indicated the instrument used (c-ToF-AMS and PILS):

*"During most of the campaign there was a reasonable agreement (slope = 1; $R^2$ = 0.6) between the PM1 $SO_4^{2-}$ (c-ToF-AMS) and the TSP $SO_4^{2-}$ (PILS) concentration, with the exception of periods of high sea salt concentrations when the TSP $SO_4^{2-}$ were significantly higher (slope = 0.5; $R^2$ = 0.2 for TSP Cl- concentrations > 10 µg $m^{-3}$; see Supplementary Figure S2)."*

**L340 low-volatility/ highly oxidized L340 Make title bold**

Authors: This has been fixed. "typical of low-volatile/highly oxidized OOA" now reads as "typical of low-volatility/highly oxidized OOA".

**L511-516 Night-time nucleation events have also been observed in the Eastern Mediterranean (Kalivitis et al. 2012)**

**Reference**
**Kalivitis, N., Stavroulas, I., Bougiatioti, A., Kouvarakis, G., Gagné, S., Manninen, H. E., Kulmala, M., and Mihalopoulos, N.: Night-time enhanced atmospheric ion concentrations in the marine boundary layer, Atmos. Chem. Phys., 12, 3627-3638, https://doi.org/10.5194/acp-12-3627-2012, 2012**

Authors: This has now been included and referenced:

*"Night-time NPF events have also been observed in the Eastern Mediterranean (Kalivitis et al., 2002)."*
* * *
**RC2**

**This study presents results on composition of fine PM fraction (approx. PM1) in Lampedusa, an island site in the southern central Mediterranean), by using a cToF-AMS. Results were obtained during the first CHARMEX Special Observation Period (SOP1) in summer 2013. Results were compared with similar studies performed in the Mediterranean region in different periods and specifically with those obtained at Ersa site, Corsica, during the sampling period. The novelty of this work lies in the fact that it is the first study of this type carried out on an island in the central Mediterranean. One of the main concerns of this study is the short duration of the sampling period (less than one month). This can affect its representativeness, the comparison with other studies and the interpretation of the results. However, results can be considered of interest in the framework of the CHARMEX SOP1.**

**Variations in PM1 composition are interpreted as a function of the origin of air masses. Higher concentrations of sulfate were obtained during transport from eastern Mediterranean, probably due to the impact of emissions from this region. A clear variation was also observed for the LO-OOA/MO-OOA ratios, with a higher contribution of the most oxidized aerosols with transport form the east.**

**Authors attempt to study the aging of aerosols during transport by comparing PM1 composition at Lampedusa and Ersa, when affected by the "same" air masses. Comparison was performed for the different clusters defined. This comparison was mainly focused on sulfate; differences were related to the accumulation of SO4 and the SO2 conversion (mainly related to the shipping emissions). A significant increase was obtained for sulfate concentrations during transport of air masses form the East. As shown in Figure 13, during transport from eastern Mediterranean, it seems that the Lampedusa site is affected by other air masses different to those impacting at Ersa. Thus, higher concentrations of sulfate at Lampedusa may be related to the impact of air masses from the East, that are not impacting at Ersa. Therefore, the proposed methodology has some limitations for estimating the aging under these scenarios.**

The authors thank the reviewer for her/his time in reviewing the manuscript and their suggestions. Many of the reviewers comments were regarding small technical/grammatical errors which have been addressed (and described in more detail below). The reviewer has two larger concerns regarding the representativeness of the study period as well as the conflation of aging processes with the possible influence of different air masses

The reviewer's first concern is regarding the short duration (less than one month) of the field campaign, although she/he acknowledges the scope of the study within the broader framework of Charmex. Due to the scale and logistics of studies such as this one with extensive instrumentation, it is extremely difficult and expensive to perform longer-term measurements. Comparable studies which are summarised in this study (see updated Figure

8) are also of a similar length, with the exception of those on continental Europe that are more easily accessible. Furthermore, while we agree that the length of the campaign period is not enough to measure longer term climatology or seasonality, the link between the aerosol composition and size and the air mass back trajectory cluster analysis provides some detail about how the wider synotic conditions could influence the aerosol composition at Lampedusa. The authors do agree though that long term (multi-seasonal or multi-year) measurements at remote sites would be very valuable.

The reviewer's second concern is regarding the comparisons between the sulphate concentrations measured concurrently at the Lampedusa and Ersa sites and the limitations of the methodology to estimate aging between the two sites. The reviewer is correct in that we do not know if there is an aging process (e.g. $SO_2 \rightarrow SO_4^{2-}$) between the two sites or if there are other air mass from the east that are impacting the sulphate concentrations. In order to address this concern, we have changed the title of the section from: *"3.6 Evidence of aging across the Mediterranean"* to *"3.6 Accumulation of sulphates across the Mediterranean"*. We have also changed instances when "growth" of sulphate to "accumulation" of sulphates which is more agnostics about the origin and mixing state of the sulphate aerosol. Lastly, we have changed the following sentence: *"It is expected that the accumulation of sulphate would increase as the total travel time increases due to the opportunity for SO2 conversion."* to: *"It is expected that the accumulation of sulphate would increase as the total travel time increases due to the opportunity for SO2 conversion or from the addition of sulphate from separate air masses which are not accounted for in the HYSPLIT model."*.

In addition to the Minor changes outlined below, several other changes in the manuscript have been made (e.g. reference formatting, typos). The major change is that data presented in Table 1 and Supplementary Table S1 has now been combined and displayed in a new figure (Figure 8; see below). This figure summarises the $PM_1$ composition from all of the previous studies around the Mediterranean basin that have performed a PMF analysis. This new Figure makes it much easier to compare the results of our study to these previous studies by visualising the composition with pie charts pointing to the sampling location, rather than presenting the data in tables. Subsequent Figure and table numbers have been updated.

[Figure]

*Figure 8 A summary of studies that have investigated NR-PM₁ composition (including PMF of OA) around the Mediterranean basin. Only studies that have investigated PMF-based OA source apportionment are reported. Pie charts display the average concentration during each study where green corresponds to organics, red to sulphates, orange to ammonium, blue to nitrate, pink to either chlorides or sea salt and black to elemental or black carbon. The OA fraction acronyms correspond to the following:HOA: Hydrocarbon-like Organic Aerosol, SV-OOA: Semi-volatile oxygenated Organic Aerosol, LV-OOA: Low-volatility oxygenated Organic Aerosol, BBOA: Biomass burning Organic Aerosol, COA: Cooking Organic Aerosol, OOA: Oxygenated Organic Aerosol, F4: "Factor -4" (unidentified PMF factor), IndOA: Industry-related Organic Aerosol, OB-OA: "Olive-branch Organic Aerosol. See Supplementary Tables S1 and S2 for further details about the sampling locations, instruments used and pie chart values. ¹This study collected on PM2.5 filters and nebulised into an HR-ToF-AMS. ²Excludes fire-periods.*

**Minor changes**

**Line 99 (e.g. FLEXPART; (Stohl et al., 2005)**

Authors: A closing bracket has been added - "(e.g.FLEXPART; (Stohl et al., 2005)" now reads "(e.g. FLEXPART; (Stohl et al., 2005))".

**Line 108: (PMF; (Paatero, 1997;**

Authors: A closing bracket has been added - "(PMF; (Paatero, 1997; Paatero and Tapper, 1994)" now reads "(PMF; (Paatero, 1997; Paatero and Tapper, 1994))"

**Lines 144-146: is the first "detailed characterization" in the Central Mediterranean, at Lampedusa, or during the CHARMEX project?**

Authors: It is the first detailed characterization of PM1 in the central remote Mediterranean in general. The sentence has been changed from:
*"In this paper, we present the first detailed characterisation of PM1 in the central Mediterranean region from measurements of size-resolved chemical composition from the island site of Lampedusa during the ChArMex/ADRIMED SOP-1a."*

and now reads

*"In this paper, we present the first detailed characterisation of PM1 in the central remote Mediterranean region, using measurements of size-resolved chemical composition on the island site of Lampedusa during the ChArMex/ADRIMED SOP-1a."*

**Line 162: Please, indicate the sampling flow**

Authors: The sample flow of the TSP (16 lpm) has now been indicated.

**Line 167: for major inorganic and organic. . .**

Authors: This has been fixed. "Samples were analysed for major and organic anions..." now reads as "Samples were analysed for major inorganic and organic anions…"

**Line 175: please indicate flow for MOUDI; did you use the same TSP inlet for all the instruments?**

Authors: The flow rate for the MOUDI cascade impactor (10 lpm) has been provided. A TSP inlet was used for the MOUDI, PILS, c-ToF-AMS and SMPS which has now been explicitly stated in the respective descriptions of each inlet.

**Line 177: which kind of filters did you use?**

Authors: PTFE (2 μm pore size) and coated with high quality vacuum grease (Dekati DS-515) to avoid bouncing. This has now been indicated.

**Line 179: Denjean et al. (2016)).**

Authors: This has been fixed. "Denjean et al., (2016)) now reads as "Denjean et al., (2016)."

**Line 214: by (Ovadnevaite et al., (2012).**

Authors: This has been fixed. "...as proposed by (Ovadnevaite et al., 2012)" now reads as "...as proposed by Ovadnevaite et al., 2012."

**Line 215: High uncertainty estimation of the sea salt**

Authors: The high uncertainty estimated of the sea salt has been explained in more detail, although it was not quantified during the experiment. The section regarding the sea salt estimation now reads :

*"The PM1 sea salt concentration was estimated in the cTof-AMS by applying a scaling factor of 102 to the ion fragment (using the cumulative peak fitting analysis) at 57.98 assigned to NaCl as proposed by Ovadnevaite et al., 2012. This scaling factor was determined by nebulising monodisperse 300 nm (mobility diameter) NaCl particles into the cToF-AMS and comparing the NaCl$^+$ signal to the total mass calculated using the number concentration from a CPC-3010. This calibration was done outside of the campaign but with similar tuning*

*conditions. The sea salt-SO4$^{2-}$ (ss-SO4$^{2-}$) was calculated as 0.252 \* 0.3 \* [seasalt], where 0.252 is the mass ratio of SO4$^{2-}$ to Na$^+$ in sea salt and 0.3 is the mass ratio of Na$^+$ to sea salt (Ghahremaninezhad et al., 2016). Given these assumptions, the uncertainty in the seasalt concentrations are likely to be significantly higher than the typical 20%, although the total contribution of seasalt to the PM$_1$ fraction was very small (0.30 μg m$^{-3}$ ; <4 %). Samples were analysed for major and organic anions"*

**Line 299: please, specify the time period of the concentrations (hourly basis; 30 minute?)**

Authors: the mean was calculated using the instrument time resolution of 3 minutes.

**Lines 312, 313, 314, 565: SO42-**

Authors: There were four instances when the sulphate anion was mistakenly written with a "+" instead of "-". These instances of SO$_4$$^{2+}$ have been changed to SO$_4$$^{2-}$.

**Line 314: (see Supplementary Figure S2).**

Authors: This has been fixed. "(see Supplementary Figure S2." now reads as (see Supplementary Figure S2)."

**Lines 319-322: this is estimation, more measurements, for a wider period are necessary for demonstrating this.**

Authors: The reviewer is correct. The wording has been altered to keep it appropriate to the scope of the study.

*"This indicates that, in these circumstances, the sea salt particles acted as a condensation sink for sulphate precursors. This has important implications for the radiative properties of these aerosols by altering the scattering properties and, potentially cloud condensation nuclei concentrations and composition."*

Has been changed to:

*"If these events are frequent, this could have important implications for the radiative properties of these aerosols by altering the scattering properties and, potentially cloud condensation nuclei concentrations and composition."*

**Lines 416-418: This comparison will depend on the sampling periods. This study was performed in summer, where high concentrations of sulfate are expected. The study by El Haddad et al (2013), also in summer, showed higher concentration of sulfate than OA**

Authors: The following sentence has been removed:

*"Furthermore, the contribution of ammonium sulphate was higher in this study than of all those undertaken in the eastern Mediterranean basin, highlighting the role contribution of sulphates across the Mediterranean."*

This original sentence was based on the reported ammonium sulphate concentrations in this study and those from around the Mediterranean basin and was supposed to point out that those in this study were higher than *western* (not eastern as initially stated) Mediterranean. Although as the reviewer points out, making this comparison depends on the sampling periods which are not always in the same season. A more detailed study of local $SO_2$ sources near each of the sites would be beneficial.

**Line 438. In figure 8: concentration of sulfate and ammonium seem higher during E – NE air masses; not north west as stated here; a similar pattern to that described for MO-OOA (Line 448).**

Authors: The reviewer is correct. "north-westerley" has been changed to "north-easterly". Furthermore, Figure 8 is now Figure 9.

**Line 450: Pattern of LO-OOA is similar to that of HOA Lines 463-466; Section 3.4. Figure 6. There is a clear difference between the ratio LO-OOA/MO-OOA during the eastern and western air masses; any comment on this?**

Authors: Although the polar-plot pattern of LO-OOA is similar to that of HOA, the time series of the two factors were not well correlated ($R^2 = 0.30$).

The potential explanation for different contributions of LO-OOA and MO-OOA during the eastern and western air masses has now been reiterated at the end of the next paragraph:

*"The higher contribution of MO-OOA compared with LO-OOA from eastern air masses, and vice-versa during western air masses, could be indicative of different OA sources prior to oxidation or due to different photochemical aging between the two directions."*

More measurements, and likely using a higher resolution AMS, would be required to distinguish smaller differences in the MO-OOA and LO-OOA and their differing contributions from the east and west. It is possible that their courses are the same or similar, however they have been kept as separate factors due to their different diurnal and temporal trends and relation to different air masses.

**Section 3.5. The measurements of size distribution are limited to the 14-600 nm fractions. The lower size is relatively high for studying the nucleation episodes. Moreover the different air humidity measured for the air clusters defined (sampling was at ambient conditions Line 469-472) may affect these measurements.**

Authors: We now indicate that how statements regarding nucleation are constrained to observations > 14 nm. We also indicate that the humidity could shift the measured size distribution to larger sizes:

*It should be noted that these size distributions are under ambient conditions without an inlet drier which could shift the size distribution to larger sizes if water is present. The ambient relative humidity for each air mass back trajectory cluster was: Eastern Mediterranean (53%), Central Europe (61%), Atlantic (74%), Western Mediterranean (70%), Mistral (low) (67%) and Mistral (high) (74%). Although the higher temperature inside the PEGASUS mobile laboratory could lower the relative humidity at the sampling point of the SMPS with respect to the ambient relative humidity, this was not measured or logged during the campaign.*

**Lines 485-487: Has the "nucleation mode ratio" been previously defined? Can you add a reference?**

Authors: It has not been previously defined. We have, however, decided to remove this term and simply state it as "the ratio of the particle number concentration between 14 - 25 nm and 14 - 600 nm" or "the ratio of sub-25 nm and sub-600 nm particle number concentrations".

**Line 501: Please, add a reference for shipping emissions**

Authors: References have now been included for the statement regarding vanadium and nickel in shipping emissions (Healy et al., 2009; Isakson et al., 2001.):

Healy, R. M., O'Connor, I. P., Hellebust, S., Allanic, A., Sodeau, J. R., & Wenger, J. C.: Characterisation of single particles from in-port ship emissions. Atmospheric Environment, 43(40), 6408-6414, 2009.

Isakson, J., Persson, T. A., & Lindgren, E. S.: Identification and assessment of ship emissions and their effects in the harbour of Göteborg, Sweden. Atmospheric Environment, *35*(21), 3659-3666, 2001

**Figure 6. caption: "less oxidized" (LO-OOA) . . .**

Authors: The caption for Figure 6 has been changed from "..."less oxidised" (OOA)" to "..."less oxidised" (LO-OOA)".